# A chemical proteomics approach for global mapping of functional lysines on cell surface of living cell

Ting Wang[1,3], Shiyun Ma[1,3], Guanghui Ji[1], Guoli Wang[2], Yang Liu[2], Lei Zhang [2], Ying Zhang [1,2] ✉ & Haojie Lu[1,2] ✉

Cell surface proteins are responsible for many crucial physiological roles, and they are also the major category of drug targets as the majority of therapeutics target membrane proteins on the surface of cells to alter cellular signaling. Despite its great significance, ligand discovery against membrane proteins has posed a great challenge mainly due to the special property of their natural habitat. Here, we design a new chemical proteomic probe OPA-S-S-alkyne that can efficiently and selectively target the lysines exposed on the cell surface and develop a chemical proteomics strategy for global analysis of surface functionality (GASF) in living cells. In total, we quantified 2639 cell surface lysines in Hela cell and several hundred residues with high reactivity were discovered, which represents the largest dataset of surface functional lysine sites to date. We discovered and validated that hyper-reactive lysine residues K382 on tyrosine kinase-like orphan receptor 2 (ROR2) and K285 on Endoglin (ENG/CD105) are at the protein interaction interface in co-crystal structures of protein complexes, emphasizing the broad potential functional consequences of cell surface lysines and GASF strategy is highly desirable for discovering new active and ligandable sites that can be functionally interrogated for drug discovery.

Membrane proteins on the cell surface play crucial physiological roles and represent a major category of drug targets, with the majority of therapeutics aiming to modify cellular signaling by targeting these proteins[1,2]. Among the small molecule drugs approved by FDA, those targeting membrane proteins account for more than 60% of all clinical drugs[3]. Despite their great significance, ligand discovery against membrane proteins has posed a major challenge mainly due to the special property of their natural habitat[4]. Compared to cytosolic proteins, membrane proteins are notoriously difficult to study because their hydrophobicity makes them lose the necessary cellular characteristics and may be inactivated following their extraction from the cell membrane. Various studies have been performed to investigate membrane proteins through labeling specific groups on the cell surface or differential centrifugation[5–9]. However, to date, no study specifically focused on the amino acid reactivity of membrane proteins has been reported. Therefore, the development of global mapping amino acid sites reactivity methods towards membrane proteins on living cells is highly desirable and could lead to the discovery of new active and ligandable sites that can be functionally interrogated for drug discovery. Chemoproteomic platforms, in particular activity-based protein profiling (ABPP), have arisen to tackle the undruggable proteome by using reactivity-based chemical probes and advanced quantitative mass spectrometry-based proteomic approaches to enable the discovery of "functional hotspots" on proteome-wide. The activity-based chemical probes with a warhead towards covalent bonding of certain amino acid types can report the activity of a

[1]Liver Cancer Institute, Zhongshan Hospital and Department of Chemistry, Fudan University, Shanghai, China. [2]Institutes of Biomedical Sciences and NHC Key Laboratory of Glycoconjugates Research, Shanghai, China. [3]These authors contributed equally: Ting Wang, Shiyun Ma. ✉e-mail: ying@fudan.edu.cn; luhaojie@fudan.edu.cn

particular site globally and provide valuable insight into the design and development of drugs or inhibitors against these biologically active sites[10,11]. The success of covalent inhibitors in the clinic has stimulated the development of chemical proteomic technologies to identify new hyper-reactive and ligandable sites for covalent targeting[12]. Therefore, beyond the successes of ABPP-based chemical proteomic approaches in ligand discovery against cysteine hotspots[13], chemical proteomic approaches have also revealed unique insights into reactivity and ligand ability targeting lysine[14,15], tyrosine[16]. Lysine shows rich chemistry through its nucleophilic amine group and is abundant in various active and allosteric sites[17]. For example, it has been shown that disease-associated protein, induced myeloid leukemia cell differentiation protein (Mcl-1), a key survival factor in a wide range of human cancers, can be inhibited through covalent modification of a lysine side chain to interrogate its function[18]. Furthermore, since the cysteine residues are typically engaged in disulfide bonds for proteins on cell surface, lysines may serve as an attractive target for covalent binding to specific cell surface proteins[19]. However, the low nucleophilicity of lysine compared to cysteine still makes selective targeting lysine a highly formidable task[20]. For example, N-Hydroxysuccinimide (NHS), as a common lysine reactive group, has been used in mapping the reactivity of nucleophilic ligandable hotspots[21] and cell surface labeling[22], but showing reactivity with other amino acids and easy to hydrolyze[23]. Moreover, current methods used for measuring amino acid reactivity, including those targeting lysine and cysteine, all lack specificity towards cell surface proteins. The method of proteome-wide study of reactive sites specifically on the cell surface has not been reported. Here, we report the development of a novel chemical proteomics strategy, termed global analysis of surface functionality (GASF), to map the cell surface functional lysine sites on living cells in a system-wide manner. To the best of our knowledge, this is the first attempt to map cell-surface lysine reactivity in living cells.

## Results

### Design and characterization of the probe

Profiling reactive lysine on living cells requires careful consideration in designing the chemical proteomics probe. First, the labeling reaction should be highly efficient and selective towards lysine while compatible with natural cellular conditions. Outside their native contexts, membrane proteins may be inactivated or lose necessary cellular features, such as ligand binding. Second, the labeled probe needs to be membrane specific, and the labeling speed should be as fast as possible in order to avoid diffusion and endocytosis of the probe into cell. Last, cell surface proteins usually exist in nanomolar to low micromolar concentration, and the poor solubility of membrane proteins makes them even easily lost during multiple sample processing steps[24]. To selectively investigate the cell surface proteins, the probe should have an isolation group for purifying the cell surface proteins for quantitative proteomics analysis. Accordingly, we designed a new chemical proteomic probe OPA-S-S-alkyne with the following functional groups (Fig. 1a). Central to this probe is (1) an electrophilic warhead for covalent binding of lysine on proteins. The covalent reaction used in this study was chosen based on ortho-phthalaldehyde (OPA) and amine conjugation to enable the highly efficient and selective lysine labeling because OPA can react chemoselectively with lysine rapidly under the physiological condition via formation of phthalimidines[23,25–27]; (2) a linker that tunes the reaction as well as minimizes undesirable labeling on inner cell proteins. Herein, a disulfide-linker is designed in this probe as it is known that the disulfide-linker helps enhance the purity of the following cell surface protein fraction[28]. The disulfide bond in the probe will be cut off by the reducing cytoplasmic environment in the cell. Therefore, even if the probe enters the cell, the proteins in the cell will not be captured during the following purification step; (3) an affinity tag for purification of labeled membrane proteins from their native environment. For this

purpose, we installed an alkyne into the probe and then used the click chemistry between the alkyne-functionalized probe and azido–biotin assisted by Cu-Catalyzed Azide–Alkyne Cycloaddition (CuAAC) to introduce a biotin tag on the labeled proteins for further streptavidin purification[29]. The synthesis of the probe is described in detail in the Supporting Information (Supplementary Fig. 1). The identity of the probe was characterized through analytical reverse phase high-performance liquid chromatography (RP-HPLC), electrospray-ionization mass spectrometry (ESI-MS), and nuclear magnetic resonance (NMR), as shown in Supplementary Fig. 2–4. ESI-MS spectra showed the m/z of $[M + H]^+$ was 407.26 (Supplementary Fig. 2), which was consistent with the theoretical value (OPA-S-S-alkyne, $C_{19}H_{22}N_2O_4S_2$). Based on the 1H and 13C NMR spectrum of OPA-S-S-alkyne in acetonitrile-$d_3$, key structural elements including the two aldehyde groups [δH 10.48 (1H, s), 10.44 (1H, s); δC 193.3, 192.9], two amide groups [δH 6.86 (1H, br s), 6.59 (1H, br s); δC 171.4, 171.1], one benzene ring [δH 7.91 (1H, d, J = 7.8 Hz), 7.82 (1H, s), 7.68 (1H, d, J = 7.8 Hz); δC 146.8, 137.0, 135.2, 134.7, 131.7, 131.7], and one terminal alkyne group [δC 80.5, 71.4], were revealed (Supplementary Fig. 3).

### Reactivity of OPA-S-S-alkyne with the model peptide and protein

We first examined the capability of this probe to react with lysine using a model peptide (Ac-GGYTLVSGYPK) and two standard proteins of bovine serum albumin (BSA) and myoglobin. The chemical reaction between the OPA group on the probe and lysine on peptides or proteins is illustrated in Fig. 1a. After incubating the probe with model peptide or standard proteins in PBS buffer at room temperature for a certain time, the reaction mixtures were quenched with 1 M glycine solution. The matrix-assisted laser desorption/ionization mass spectrometry (MALDI MS) spectra confirmed the successful labeling of model peptide by observation of the mass increase of 388.09 Da (Fig. 1b). The reaction rate assay further showed that the probe can efficiently conjugate to the lysine residue on model peptide within 1 min (Fig. 1c). Then we investigated the capability of this probe to label lysine in proteins and labeling of the OPA-S-S-alkyne probe on multiple lysines in myoglobin was observed (Fig. 1d). The efficiency of conjugating the probe to proteins over labeling time and probe concentration was also investigated. To visualize the probe-labeled proteins on gel, the labeled BSA was also clicked with a fluorogenic azide tag before being analyzed by sodium dodecyl sulfate-polyacrylamide polyacrylamide gel electrophoresis (SDS-PAGE) and then detected by in-gel fluorescence scanning. The fluorescent gel showed that the BSA protein was labeled by the OPA-S-S-alkyne probe in a time and concentration-dependent manner and the probe can also label standard protein in a short time (Fig. 1e). The probe concentration of 10 μM is enough for the protein labeling[27]. Moreover, as we designed a disulfide in the linker, the disulfide bond is expected to be cleaved once the probe is placed in a reducing environment. To confirm this hypothesis, we added dithiothreitol (DTT) to reduce the disulfide bond of the probe-tagged peptide and compared the MALDI MS spectra before and after adding DTT. As expected, the disulfide bond was cleaved almost completely by DTT (Supplementary Fig. 5).

### Development of GASF workflow on living cells

With evidence in hand that the probe can label lysine with high efficiency, we next used the probe to label Hela cells in vivo and examined the labeling performance on living cells (Fig. 2a). After washing away the cell culture media by PBS three times, Hela cells were incubated with the probe and then quenched the labeling using 1 M glycine. Then, the cells were washed by PBS again to remove the excess probe and collected for cell lysis. The conjugation efficiency over labeling time was determined by CuAAC to a fluorogenic azide tag followed by SDS–PAGE separation and in-gel fluorescence scanning. The fluorescent signal showed that the probe can also label the cell in vivo very fast (<1 min) (Fig. 2b). Then, the cells with or without labeling were

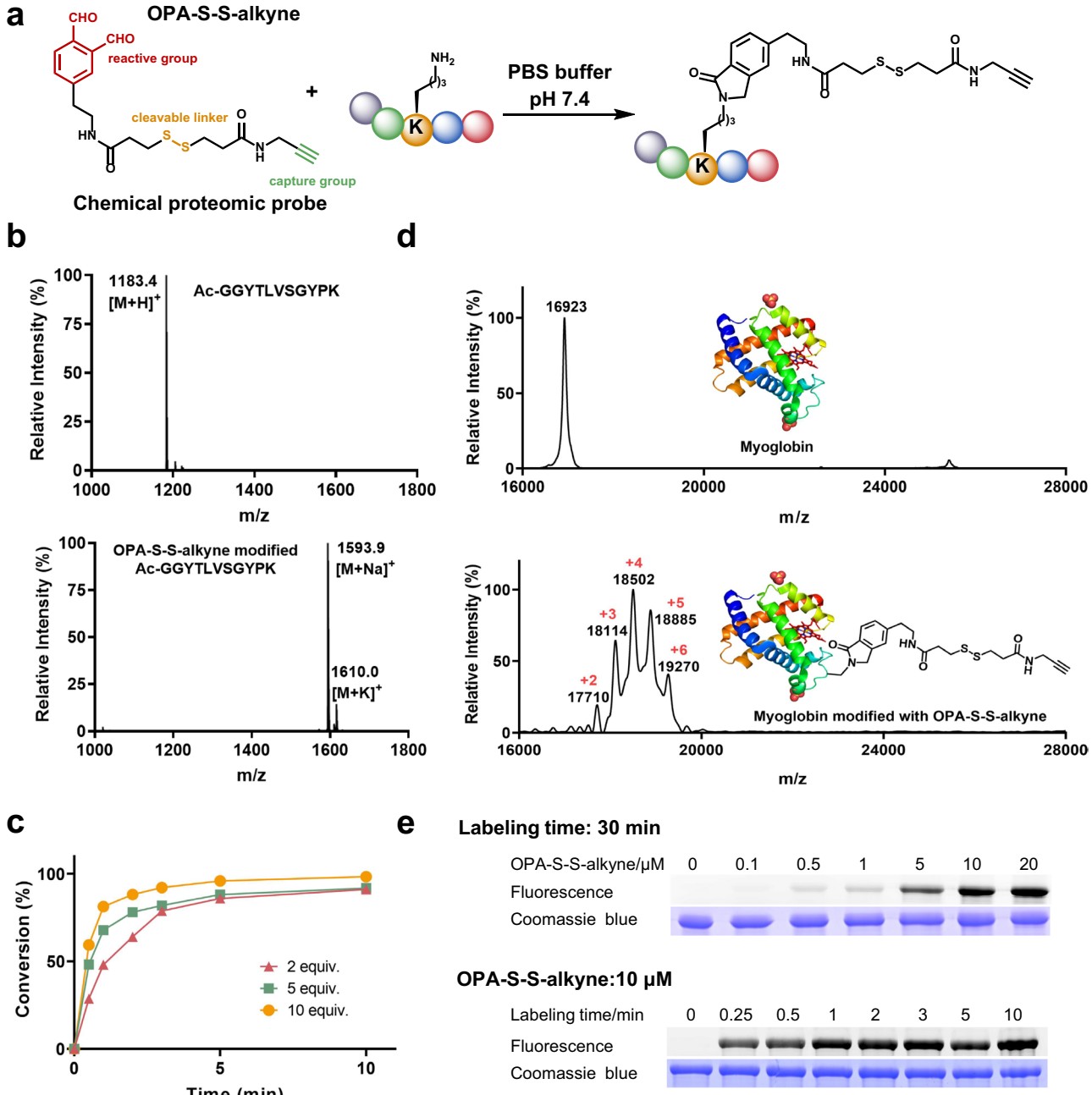

**Fig. 1 | The trifunctional reductant-cleavable chemical proteomic probe OPA-S-S-alkyne for fast labeling of lysine. a** The chemical structure of OPA-S-S-alkyne and chemical reaction between the probe and lysine on peptides or proteins in PBS buffer (pH = 7.4). **b** MALDI MS spectra of the model peptide (Ac-GGYTLVSGYPK) before (top) and after (bottom) reaction with OPA-S-S-alkyne. **c** Reaction rate assay of model peptide (Ac-GGYTLVSGYPK) with different equivalent OPA-S-S-alkyne over time (0.5 min, 1 min, 2 min, 3 min, 5 min, and 10 min). **d** MALDI MS spectra of myoglobin protein before (top) and after (bottom) modified with OPA-S-S-alkyne. The numbers marked in red represent how many lysines of myoglobin are modified by OPA-S-S-alkyne. **e** In-gel fluorescence scanning of the protein covalent labeled by OPA-S-S-alkyne followed by click with CalFluor 488 Azide. Top: Concentration-dependent labeling of BSA protein for 30 min; Bottom: Time-dependent labeling of BSA protein (10 μM). Images shown are representative of three independent experiments. Source data are provided as a Source Data file.

fixed and conjugated to CalFluor 488 Azide by CuAAC, and analyzed by confocal imaging. The fluorescent signal was only observed on the cell surface and extracellular space of the cell (Fig. 2c, left). Meanwhile, no fluorescence in the negative group was observed, which indicates this probe could efficiently and selectively conjugate to the cell surface proteins (Fig. 2c, right), whereas OPA-alkyne labeled both cell surface and intracellular proteins (Supplementary Fig. 6). SDS-PAGE of soluble and particulate proteomes also showed that OPA-S-S-alkyne mainly labeled membrane proteins (Supplementary Fig. 7). In addition, GSH, the major contributor to the intracellular reducing environment, could

cleave the disulfide in a concentration dependent manner (Supplementary Fig. 8), which may enhance the OPA-S-S-alkyne selectivity of cell surface protein. At the same time, we tested whether the probe affected the cell viability with the WST-1 assay. The cell viability had no significant change after 10 min treatment of the probe (Fig. 2d). We also treated the cells with different concentrations of OPA-S-S-alkyne and incubated with STYOX Green and we did not see an obvious change of the membrane permeation after probe labeling (Supplementary Fig. 9). These results indicated that the probe was suitable for in vivo labeling. To further demonstrate that the labeling occurs on the

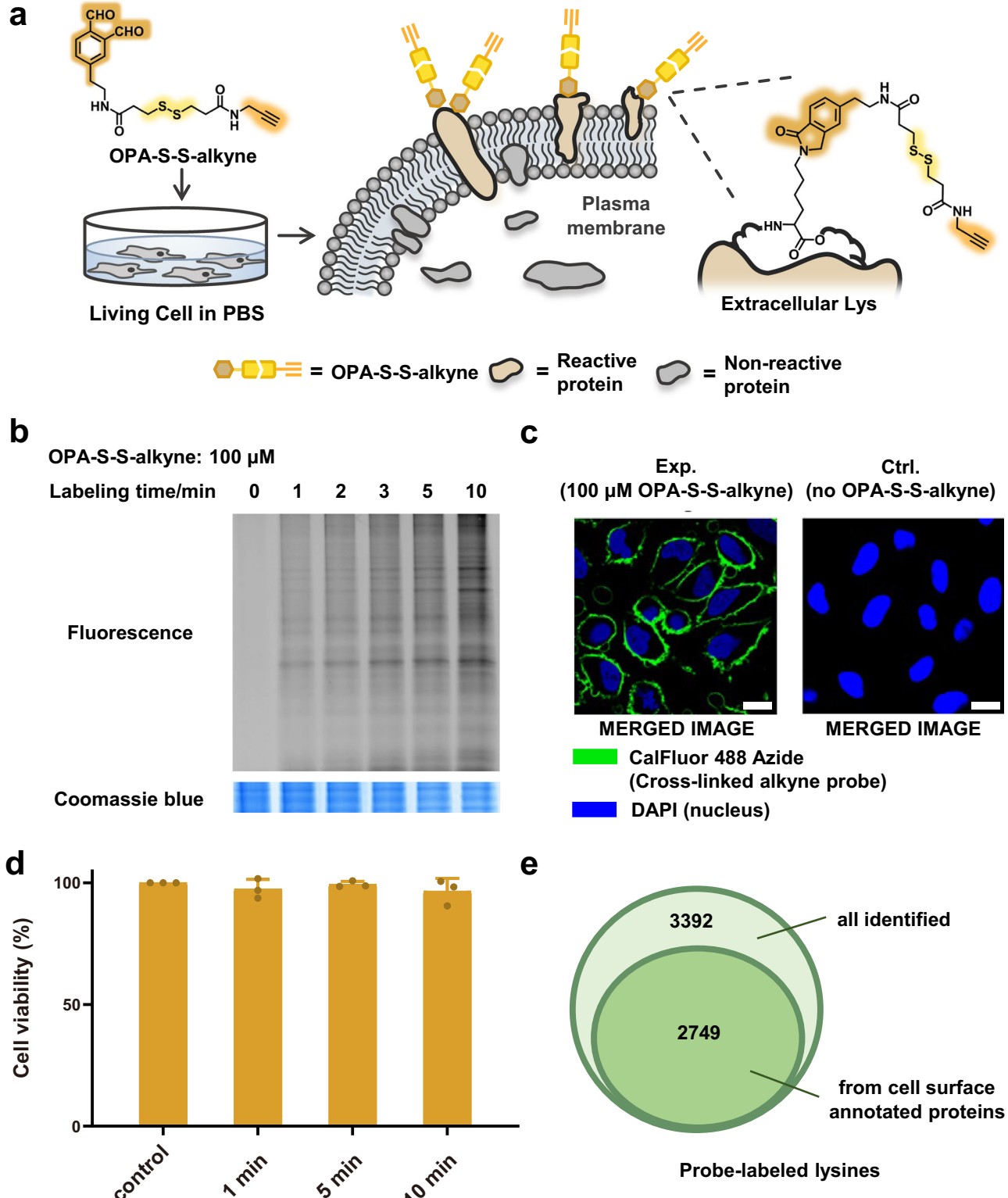

**Fig. 2 | OPA-S-S-alkyne enables the efficient and selective targeting cell surface lysines on living cells. a** Scheme for labeling cell surface lysines by OPA-S-S-alkyne on living cells. **b** The time for efficient labeling was evaluated by labeling Hela cells with OPA-S-S-alkyne (100 μM) for different time followed by protein extraction and click with CalFluor 488 Azide, then analyzed by SDS–PAGE and in-gel fluorescence scanning. Images shown are representative of three independent experiments. **c** Living Hela cells were labeled by OPA-S-S-alkyne (100 μM) followed by click with CalFluor 488 Azide and analyzed by confocal fluorescent microscopy. Images shown are representative of three independent experiments. Scale bars, 40 μm.

**d** WST-1 cell viability assay of living cells labeled within 10 min. PBS-treated cells were used as positive control. Quantitative data from three independent experiments are shown as the mean ± SD. **e** Single run LC-MS/MS analysis of probe-labeled lysines. Living Hela cells were labeled with OPA-S-S-alkyne (100 μM) followed by protein extraction, reacted with azide-biotin, digested by trypsin, enriched by avidin beads, eluted by DTT/IAA and analyzed by LC-MS/MS. A total of 3392 probe-labeled lysines were identified, among them 2749 lysines were from cell surface annotated proteins. Source data are provided as a Source Data file.

cell surface, we performed proteomics analysis to identify these proteins. The probe-labeled cells were lysated and the labeled proteins were conjugated to azide-PEG$_4$-biotin to introduce a biotin tag on the lysines. After digestion, the peptides were captured on avidin beads, and eluted by dithiothreitol (DTT)/iodoacetamide (IAA), followed by identification using liquid chromatography tandem mass spectrometry (LC–MS/MS). In total, 3392 probe-labeled lysines were identified, and 2749 lysines among them (~81% of the total identified labeled lysines) were from cell surface annotated proteins by gene ontology (GO) annotation (Fig. 2e). Another cell line, HEK293T was also labeled and enriched following the same workflow, showing similar selectivity and GO annotation (Supplementary Fig. 10), indicating that GASF workflow was applicable to various cell lines. Then we further examined the distribution of probe labeled lysines of transmembrane proteins (TMPs) according to their topological domain (extracellular/transmembrane/cytoplasmic) in Uniprot database. Several typical TMPs including epidermal growth factor receptor (EGFR) and transferrin receptor protein 1 (TFR1) with a single transmembrane domain (TMD), and highly hydrophobic TMPs, like adhesion G protein-coupled receptor E5 (CD97) and Frizzled-2 (FZD2) with multiple TMDs were analyzed. All probe labeled lysines of these TMPs were located in the extracellular region, further indicating that this probe was highly selective towards cell surface lysines (Supplementary Fig. 11).

## Quantitative profiling of cell surface lysine reactivity

Having demonstrated that the probe can label the cell surface lysine with high efficiency and selectivity, we next switched our focus to map the intrinsic reactivity of lysine residues on the cell surface by GASF. We applied a quantitative stable isotope labeling by amino acids in cell culture (SILAC)-based proteomic experiment to measure the intensities ratio of labeled lysines at low versus high concentrations of OPA-S-S-alkyne probe treated cells (SILAC-ABPP) (Fig. 3a). Highly reactive lysines are expected to show nearly equivalent labeling intensities at low versus high concentrations of probe, while less reactive lysines displaying concentration-dependent increases in labeling intensity. In brief, Hela cells grown in culture medium containing isotopically labeled "heavy" lysine and arginine (K8R10) or standard "light" medium (K0R0) were treated with low or high concentrations of probe OPA-S-S-alkyne (50 versus 500 μM), collected and lysed. Heavy- and light-labeled cell lysates were combined at a 1:1 ratio followed by reaction with azide-PEG$_4$-biotin, trypsin digestion, streptavidin enrichment, and eluted by DTT/IAA. The resulting peptides were analyzed by LC-MS/MS and the lysine ratio was then quantified as the median ratio across all peptides assigned to a specific lysine, wherein high, medium and low reactivity lysines were distinguished by their respective SILAC ratio values ($R_{10:1} < 2$, $2 < R_{10:1} < 5$, $R_{10:1} > 5$, respectively)[14,16,30]. We first tested the quantitative accuracy of GASF. To do so, both the heavy and light cells were treated with 500 μM OPA-S-S-alkyne probe and analyzed by GASF strategy. The distribution of lysine ratios had a mean of 1.01, accurately matching the theoretical value (Supplementary Fig. 12). Then, we applied GASF to map the reactivity of lysine residues on the cell surface in living cells. In the standard "forward" SILAC experiment, the light cells were treated with 50 μM OPA-S-S-alkyne, and the heavy cells were treated with 500 μM OPA-S-S-alkyne, and we also performed a replicate "reverse" SILAC experiment, where the labeling order was switched (heavy with 50 μM and light with 500 μM OPA-S-S-alkyne) to increase the robustness and accuracy for quantitation ($n = 3$ per group).

In total, 2687 lysine residues assigned to 831 cell surface annotated proteins were quantified across 6 replicate experiments. Among these 831 proteins, 216 known TMPs have topological domain information in Uniprot database. From the 216 TMPs, 718 lysines were quantified, of which 656 lysines corresponded to 197 TMPs were located in the extracellular region (Supplementary Data 2). The remaining 48 lysines were located in the cytoplasmic or

transmembrane region and 14 lysines had no location information (Fig. 3b). Therefore, the proportion of extracellular lysines we identified was about 91%. To the best of our knowledge, there have been no studies on lysine activity on the surface of living cells, and little information is available on cell surface lysine reactivity in the existing studies. In recent work, global profiling of lysine reactivity in the human proteome, based on cell lysate, has been performed[14]. In their whole proteome data, only 18 lysines assigned to 15 TMPs were quantified, of which 8 lysines (44%) located on the cell surface, and the remaining 10 lysines were located in the cytoplasmic region (Fig. 3b). This comparison further indicated that GASF was highly specific for lysines exposed on the cell surface.

## Functional analysis of hyper-reactive lysines

After excluding these known intracellular lysines, 2639 lysine residues definitely (located in the extracellular region) and potentially (had no known location information) exposed on the cell surface were included for the following intrinsic reactivity analysis (Supplementary Fig. 13 and Supplementary Data 1). Of the 2639 lysines we quantified, the majority showed strong concentration-dependent increases in reaction with OPA-S-S-alkyne probe, indicative of residues with low intrinsic reactivity (Fig. 3c). In contrast, a small fraction of the quantified lysines exhibited hyper-reactivity with OPA-S-S-alkyne probe ($R_{10:1} < 2$) (Fig. 3c). Most proteins contained only one hyper-reactive lysine among several quantified lysines (Fig. 3d), and the atypical hyper-reactivity of these lysines was further supported by comparing their $R_{10:1}$ value to those of other lysines quantified on the same protein (Supplementary Fig. 14). The primary sequence surrounding hyper-reactive lysines did not show obvious conserved motifs (Supplementary Fig. 15), indicating that the reactivity of lysines has little association with their amino acid sequence. The functional classes of TMPs that contain hyper-reactive lysines mainly included transmembrane signal receptor, cell adhesion molecule and protein modifying enzyme (Fig. 3e and Supplementary Data 3), consistent with GO-term enrichment analysis for these proteins (Supplementary Fig. 16). 42% (94/224) of the highly reactive lysines we identified were located in the various protein domains (Supplementary Fig. 17), and nearly half (108/224) of the highly reactive lysines were annotated to be PTM sites[31] (Supplementary Fig. 18). We also found that several hyper-reactive lysines were directly involved in the formation of protein complex, such as K27 in S1PR3–Gi–scFv16 complex[32], K362 in DSG2-Had3K complex[33], K748 in ABCB1–UIC2 complex[34] and K600 in Met-InlB complex[35], displaying important roles in lipid regulation, virus infection, drug transportation, and cell migration, respectively. Moreover, for the 127 transmembrane proteins identified with hyper-reactive lysines, 31% (40/127) of them were included in the DrugBank Database, showing a tight connection with membrane-related function (Supplementary Fig. 19). All these functional analyses show that the hyper-reactive lysines we identified do bear important functions in cellular activities and have good potential to serve as drug targets.

Besides common drug targets and ligands in drug bank such as EGFR and TFR1 were identified, the receptor tyrosine kinase-like orphan receptor 2 (ROR2) and endoglin (ENG/CD105) were newly identified containing hyper-reactive lysines. The distribution of lysines of ROR2 and ENG was shown in Supplementary Fig. 20. ROR2 and ENG/CD105 have 10 and 20 lysines in their extracellular region respectively and all quantified lysines by GASF strategy were located in the extracellular region. ROR2, a receptor tyrosine kinase, is involved in the WNT signaling pathway when associated with its ligand WNT5A and facilitates polarization of cells during embryonic development. It is upregulated in a diverse set of hematologic and solid malignancies and represents as a candidate antigen for antibody-based cancer therapy[36]. Recently, when developing therapeutic antibody against ROR2, the co-crystallation analysis shows that the K382 of ROR2 is at the interface between ROR2 and antibody, and forms hydrogen bond interaction

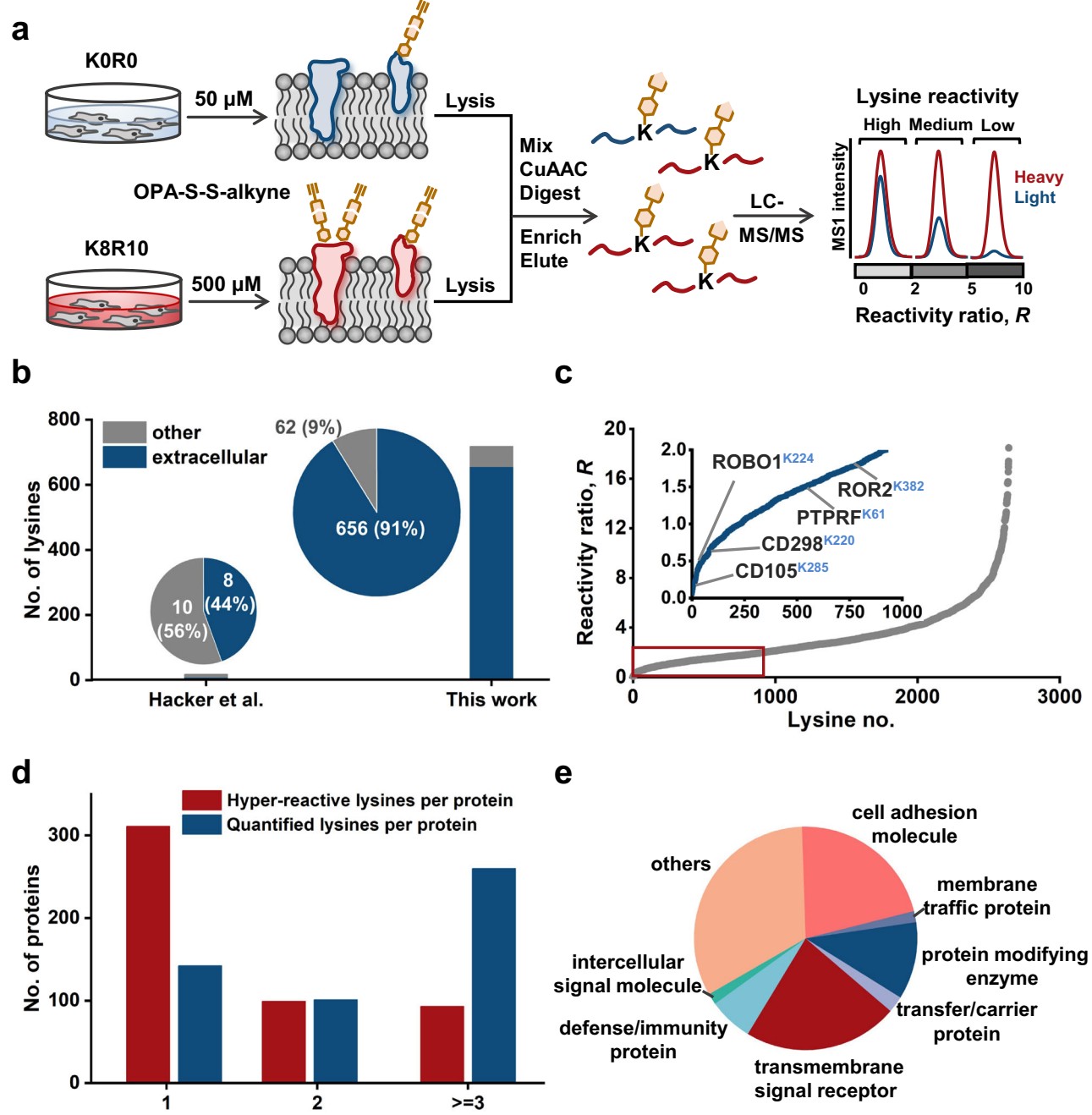

**Fig. 3 | The GASF strategy for proteome-wide quantification of cell surface lysine reactivity. a** Schematic illustration of the cell surface lysine reactivity profiling workflow. **b** Comparison of the cell surface lysines of TMPs (known topology domain information in Uniprot database) in our surfaceome data with previous whole proteome data.6 **c** Distribution of the SILAC-ABPP ratios for 2639 lysines definitely and potentially exposed on the cell surface from living Hela cells treated with 50 versus 500 µM OPA-S-S-alkyne. The inset shows the zoom-in view of the group of hyper-reactive lysines ($R \leq 2.0$, red boxed) with representative sites marked (K285 of CD105, K224 of ROBO1, K220 of CD298, K61 of PTPRF and K382 of ROR2). **d** Number of hyper-reactive and quantified lysines per protein shown for proteins found to contain at least one hyper-reactive lysine. **e** Distribution of functional classes of TMPs that contain hyper-reactive lysines.

with Ala-53 and Asn-55 of monoclonal antibody, further suggesting the critical role of K382 (Fig. 4a, top)[37]. Another example, ENG/CD105, a key player in angiogenesis and vascular homeostasis, is mutated in the genetic disorder HHT1 and implicated in tumor angiogenesis[38] and preeclampsia[39]. The crystal structure of the ENG in complex with its ligand bone morphogenetic protein 9 (BMP9) shows that the interface between ENG and BMP9 contains residues mutated in HHT1 and overlaps with the epitope of tumor-suppressing anti-ENG monoclonal TRC105[40]. In our case, the residue K285 of ENG identified as hyper reactive site is also at the interface (within 4.5 Å of BMP9), further

suggesting the critical role of K285 (Fig. 4a, bottom). The structural information further demonstrated the reliability of hyper-reactivity lysines we quantified and emphasized the broad potential functional consequences of cell surface lysines. Finally, we confirmed their lysine reactivity determinations by recombinantly expressing wild-type (WT) and lysine-to-arginine mutant proteins and comparing their reactivity by gel-based ABPP. HEK293T cells expressing these two proteins and their corresponding lysine-to-arginine (K−R) mutant as Flag epitope-tagged proteins were treated with OPA-S-S-alkyne probe, then probe-labeled proteins were enriched and the labeling signal was reflected by

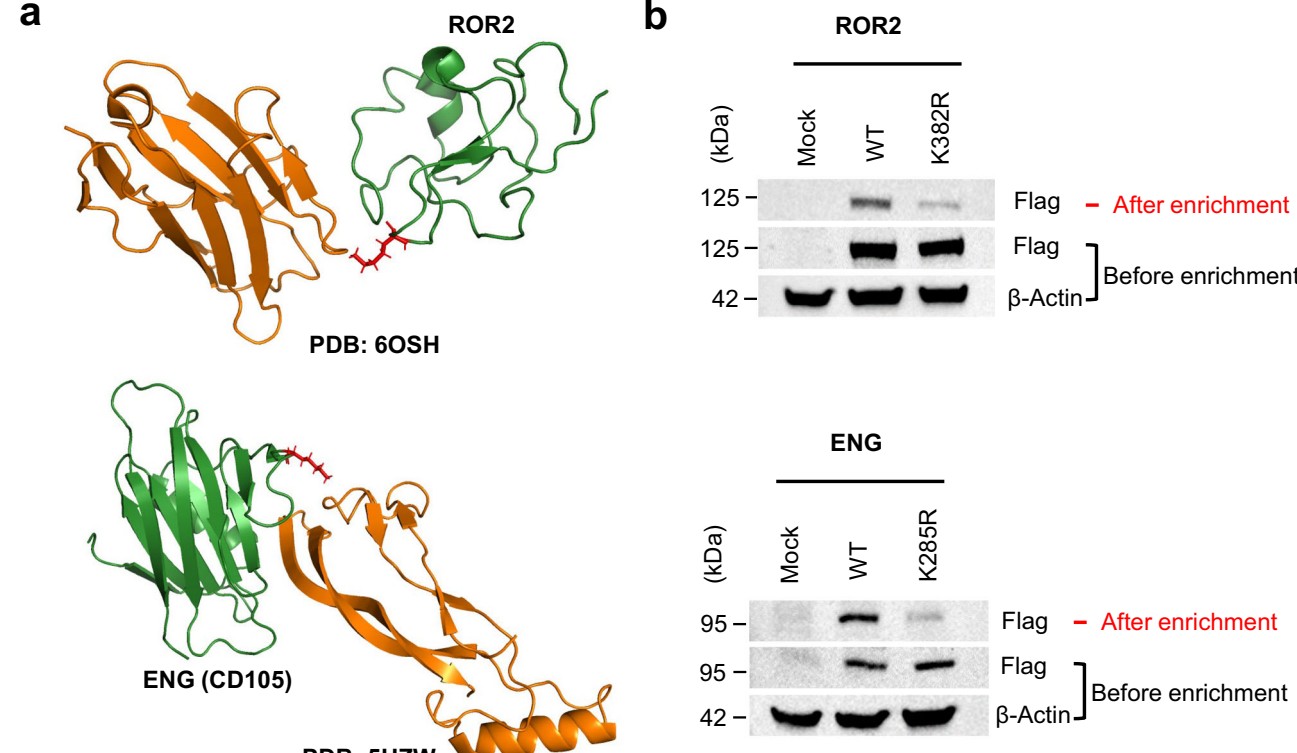

**Fig. 4 | Functional characterization of the hyper-reactive lysines in ROR2 and ENG. a** Co-crystal structures of proteins with hyper-reactive lysines in complexes with their interacting proteins. Hyper-reactive lysines (K382 for ROR2 and K285 for ENG) are shown in red, ROR2 and ENG in green, their interacting proteins in orange (parental rabbit mAb for ROR2, and BMP9 for ENG). **b** Hyper-reactive lysines can be site-selectively labeled by OPA-S-S-alkyne, the labeling was blocked by mutation of the hyper-reactive lysine to arginine. HEK293T cells expressing ROR2 and ENG (and their corresponding lysine-to-arginine mutant) as Flag epitope-tagged proteins were treated with the OPA-S-S-alkyne, reacted with azide-biotin for enrichment, eluted and analyzed by western blotting using Flag antibody. Images shown are representative of three independent experiments. Source data are provided as a Source Data file.

flag signal. Compared to WT proteins, the flag signal intensity of K−R mutant proteins was weaker, indicating that OPA-S-S-alkyne labeling of ROR2 and ENG was blocked by mutation of the hyper-reactive lysine to arginine (Fig. 4b and Supplementary Fig. 21). Among them, we further studied the role of K285 in the interaction of ENG and BMP9. And we observed that the amount of BMP9 displayed in a concentration-dependent manner in WT, whereas there was no significant change in the K285R variant (Supplementary Fig. 22), indicating that the combination of the probe and K285 blocked the interaction of ENG and BMP9. The lower amount of BMP9 in K285R mutant than WT without the treatment of our probe may be caused by the weaker interaction of K285R-ENG and BMP9 (Supplementary Fig. 23). In summary, the hyper-reactive residues identified by our strategy played an important role in the protein interaction and could be efficiently blocked by our probe, which may offer a new perspective on the drug development of concerning diseases.

## Discussion

Membrane proteins play a crucial role in biological systems and serve as targets for many drugs. However, the expression, purification, and maintenance of their active state pose significant challenges, making the design of small-molecule covalent drugs targeting membrane proteins particularly difficult. In this work, a new chemical proteomics probe for chemo-selectively labeling lysine residues on the cell surface was tailor-made and a chemical proteomic methodology GASF to globally map the reactivity and functionality of cell surface lysines in living cells was presented. Through this method, we reported the first dataset of cell surface lysine reactivity, which is almost unavailable in the previous dataset. We successfully identified numerous hyper-reactive lysines from cell surface, which may serve as potential therapeutic targets. This newly developed strategy provides an effective way to find new cell surface protein ligandable sites, which is insightful for designing inhibitors or drugs towards these proteins. Moreover, the chemical proteomic strategy is general and can be applicable to discover a variety of ligandable amino acids on cell surface by modifying the warhead of the chemical proteomic probe with different functional groups towards these amino acids.

## Methods

### Materials

Acetonitrile was obtained from Merck (Darmstadt, Germany). Trypsin was purchased from Beijing Shengxia Proteins Scientific Ltd. (Beijing, China. Bicinchoninic acid (BCA) protein assay kit was obtained from Pierce (Thermo, U.S.A.). 3-[4-({bis[(1-tert-butyl-1H-1,2,3-triazol-4-yl) methyl] amino} methyl)-1H-1,2,3-triazol-1yl] propanol (BTTP), CalFluor 488 Azide and azide-PEG4-biotin were purchased from Click Chemistry Tools (Scottsdale, U.S.A.). Neutravidin beads were purchased from Thermo Fisher Scientific (CA, U.S.A.). The other chemical reagents unless specified otherwise noted were from Sigma-Aldrich (St. Louis, U.S.A.). Distilled water was purified by a Milli-Q system (Milford, MA, U.S.A.). Model peptide (Ac-GGYTLVSGYPK) was synthesized by China Peptides Co., Ltd. (Shanghai, China).

### Synthesis of N-(3,4-diformylphenethyl)-3-((3-oxo-3-(prop-2-yn-1-ylamino) propyl) disulfaneyl) propenamide (OPA-S-S-alkyne)

The probe was designed by the authors and the synthesis was custom made from Serinno Holdings Limited. This probe will be commercially available after this work is disclosed. ${}^1$H NMR (500 MHz, CD$_3$CN)

$\delta$ = 10.48 (1H, s), 10.44 (1H, s), 7.91 (1H, d, $J$ = 7.8 Hz), 7.82 (1H, s), 7.68 (1H, d, $J$ = 7.8 Hz), 6.86 (1H, br s), 6.59 (1H, br s). $^{13}$C NMR (100 MHz, CD$_3$CN) $\delta$ = 193.3, 192.9, 171.4, 171.1, 146.8, 137.0, 135.2, 134.7, 131.7, 131.7, 80.5, 71.4. Analytical reversed-phase high-performance liquid chromatography (RP-HPLC) separations involving a mobile phase of 0.1% trifluoroacetic acid (TFA) (v/v) in acetonitrile (Solvent A) and 0.1% TFA (v/v) in water (Solvent B) were performed on a Waters UPLC H-class system equipped with an ACQUITY UPLC photodiode array detector and a Waters SQ Detector 2 mass spectrometer using a Waters ACQUITY BEH C18 column (1.7 μm, 130 Å, 2.1 ×50 mm) at a flow rate of 0.4 mL/min. The calculated purity was >95%. ESI–MS (m/z): calculated for C$_{19}$H$_{22}$N$_2$O$_4$S$_2$ [M + H]$^+$: 407.53; found [M + H]$^+$: 407.26; [M+Na]$^+$: 429.51; found [M+Na]$^+$: 429.27; [M+Na+H$_2$O]$^+$: 447.52; found [M+Na+H$_2$O]$^+$: 447.30.

## Reaction rate assay

For all experiments, N-terminal acetylated peptide Ac-GGYTLVSGYPK was dissolved in PBS buffer (pH = 7.4) to have a final reaction concentration of 0.1 mM. OPA-S-S-alkyne was added to have a final reaction concentration of 0.2 mM, 0.5 mM, and 1 mM. The reaction was stirred at room temperature. The reaction was monitored at 0.5 min, 1 min, 2 min, 3 min, 5 min, and 10 min. At each time point, the reaction was quenched by adding 1 M glycine. The conversation was monitored by MALDI MS (Rapiflex, Bruker) and the conversation percentage was calculated based on the consumption of peptide to the final product.

## MALDI-TOF analysis

Myoglobin was dissolved in PBS buffer (pH = 7.4) at the concentration of 1 mg/mL. The OPA-S-S-alkyne probe (5 eq.) was added and incubated for 30 min at room temperature. After 30 min, the reaction was quenched by adding 1 M glycine. The sample solution after desalting was ready for MALDI-TOF analysis (Rapiflex, Bruker).

## Conjugation of BSA with OPA-S-S-alkyne

Bovine serum albumin was dissolved in PBS buffer (pH = 7.4) at the concentration of 1 mg/mL (~16 μM). Then designated concentrations (0.1 μM, 0.5 μM, 1 μM, 5 μM, 10 μM and 20 μM) of OPA-S-S-alkyne probe were added and incubated for designated time (0.25 min, 0.5 min, 1 min, 2 min, 3 min, 5 min, 10 min, and 30 min) at room temperature. After incubation, the reactions were quenched by adding 4 volumes of ice-cold acetone to precipitate proteins. Then the mixture was placed at −20 °C overnight and centrifuged at 15,000 g for 30 min at 4 °C. The supernatant was discarded and the pellet was washed with ice-cold methanol twice and air-dried for 10 min. Then the proteins were incubated in DPBS buffer containing 100 μM CalFluor 488 Azide, 2.5 mM freshly prepared sodium ascorbate, 500 μM 3-[4-({bis[(1-tert-butyl-1H-1,2,3-triazol-4-yl) methyl] amino} methyl)−1H-1,2,3-triazol-1yl] propanol (BTTP), 250 μM CuSO$_4$ for 1 h at room temperature with rotation in the dark. The reacted samples were boiled for 5 min at 97 °C in 1× loading buffer, and resolved by 10% SDS/PAGE. The labeled proteins were visualized by scanning the gel on a Typhoon FLA 9500 Variable Mode Imager (GE Healthcare) using a 473 nm for excitation. The gels were then stained by Coomassie brilliant blue to show equal loading.

## Cell culture

Hela cells and HEK293T were acquired from China National Collection of Authenticated Cell Cultures. Cells were cultured in Dulbecco's modified Eagle's medium (DMEM) supplemented with 10% (v/v) fetal bovine serum (FBS) and 1% (v/v) penicillin/streptomycin in a humidified atmosphere at 37 °C with 5% CO$_2$. For SILAC experiments, Hela cells were passaged seven times in DMEM minus L-lysine and L-arginine supplemented with 10% (v/v) dialyzed FBS, 1% (v/v) penicillin/streptomycin, and 100 μg/mL regular L-arginine-HCl and L-lysine-HCl or [$^{13}$C$_6$$^{15}$,N$_4$] L-arginine-HCl and [$^{13}$C$_6$$^{15}$,N$_2$] L-lysine-HCl. The resulting

"light" and "heavy" Hela cells were then used for SILAC-based experiments.

## Living cells labeling with OPA-S-S-alkyne probe and cell lysis

Hela cells or HEK293T cells at about 80% confluency were subjected to medium removal and three washes with room-temperature PBS (pH 7.4), followed by the addition of room-temperature PBS containing OPA-S-S-alkyne. The labeling reaction was gently agitated on an orbital shaker at room temperature and then quenched using 1 M glycine. The labeled cells were washed three times with ice-cold PBS and then harvested by scraping on ice. Cell pellets were resuspended in lysis buffer containing 4% SDS, 1% (v/v) protease inhibitor cocktail (EDTA-free, Roche Diagnostics) in DPBS and sonicated for 3 min and centrifuged at 18,000 × $g$ for 10 min at 4 °C to collect the supernatant. The concentration of proteins was determined using BCA assay.

## In-gel fluorescence scanning

To evaluate the labeling efficiency, the labeling was performed using 100 μM OPA-S-S-alkyne for predetermined time points (1 min, 2 min, 3 min, 5 min, and 10 min) at room temperature in living Hela cells. Cells were lysed and protein concentration was determined as described above. The weight of 30 μg lysates were clicked with CalFluor 488 Azide and OPA-S-S-alkyne labeled proteins were visualized by SDS-PAGE and in-gel fluorescence scanning as described above. The gels were stained by Coomassie brilliant blue to show equal loading.

To evaluate the cell surface protein selectivity of OPA-S-S-alkyne, Hela cells were labeled with OPA-S-S-alkyne and were resuspended in PBS and lysed by 3× periodic 5 s ON/OFF ultrasonication followed by centrifugation at 12,000 × $g$ during 30 min. The supernatant was regarded as the cytosolic fraction, while the palleted was regarded as the membrane fraction and resuspended in 4% SDS in PBS. And the whole cell lysate was collected using 1% SDS in PBS. Protein concentration was determined as described above. Same weight of the fractions were clicked with TAMRA Azide and analyzed by SDS–PAGE and in-gel fluorescence scanning.

To test the concentration of GSH on the efficiency of disulfide cleavage in the linker, OPA-S-S-alkyne labeled BSA was mixed with different concentrations of GSH for 10 min and quenched by chloroform-methanol precipitation. Then protein pellets were resuspended in 0.4% SDS in PBS and clicked with TAMRA Azide. Proteins were visualized by SDS-PAGE and in-gel fluorescence scanning as described above.

## Cell viability assay

To evaluate the cell viability, the labeling was performed using 100 μM OPA-S-S-alkyne for 1 min, 5 min, and 10 min at room temperature in living Hela cells. The PBS-treated (no OPA-S-S-alkyne) cells were used as positive control. The cell viability was assessed using the WST1 (water-soluble tetrazolium salt-1) (ab65473; Abcam) assay according to the manufacturer's instructions.

## Membrane permeation assay

Hela cells were seeded on to 35 mm glass bottom wells and grew to about 80% confluency. Living cells were then labeled with different concentrations of OPA-S-S-alkyne or PBS or methanol (dead cells for positive control) for 10 min. After washed three times with PBS, the cells were incubated with STYOX Green for 30 min at room temperature. Living cells were analyzed using the Laser Scanning Confocal Microscope. Hela cells were seeded onto six-well plates containing glass coverslips. After growing to about 80% confluency, living cells on coverslips were labeled by OPA-S-S-alkyne as described above. The cells were washed three times with PBS, followed by incubation for 5 min in PBS containing 100 μM CalFluor 488 Azide, 2.5 mM sodium ascorbate, 500 μM BTTP, 250 μM CuSO$_4$ at room temperature in the dark. After three washes with PBS, the cells were fixed with 4%

formaldehyde in PBS at room temperature for 15 min. Subsequently, after three washes with PBS, the cells were then incubated with 1 µg/ml DAPI (4,6-diamidino-2-phenylindole) at room temperature for 1 h.

For the comparison of OPA-S-S-alkyne and OPA-alkyne labeling, after the OPA-S-S-alkyne or OPA-alkyne procedure, the cells were fixed with cold methanol at 4 °C for 15 min. Then the cells were washed three times with PBS followed by the permeabilization with 0.2% Triton X-100 in PBS for 10 min at room temperature. The cells were washed three times with PBS, followed by incubation for 2 h in PBS containing 100 µM TAMRA Azide, 2.5 mM sodium ascorbate, 500 µM BTTP, 250 µM CuSO4 at room temperature in the dark. Subsequently, after three washes with PBS, the cells were then incubated with 1 µg/ml DAPI at room temperature for 1 h.

The cells were analyzed using the Laser Scanning Confocal Microscope (Leica SP8 LSCM).

## Immunofluorescence assay

Hela cells were seeded onto six-well plates containing glass coverslips. After growing to about 80% confluency, living cells on coverslips were labeled by OPA-S-S-alkyne as described above. The cells were washed three times with PBS, followed by incubation for 5 min in PBS containing 100 µM CalFluor 488 Azide, 2.5 mM sodium ascorbate, 500 µM BTTP, 250 µM CuSO$_4$ at room temperature in the dark. After three washes with PBS, the cells were fixed with 4% formaldehyde in PBS at room temperature for 15 min. Subsequently, after three washes with PBS, the cells were then incubated with 1 µg/ml DAPI (4′,6-diamidino-2-phenylindole) at room temperature for 1 h.

For the comparison of OPA-S-S-alkyne and OPA-alkyne labeling, after the OPA-S-S-alkyne or OPA-alkyne procedure, the cells were fixed with cold methanol at 4 °C for 15 min. Then the cells were washed three times with PBS followed by the permeabilization with 0.2% Triton X-100 in PBS for 10 min at room temperature. The cells were washed three times with PBS, followed by incubation for 2 h in PBS containing 100 µM TAMRA Azide, 2.5 mM sodium ascorbate, 500 µM BTTP, and 250 µM CuSO4 at room temperature in the dark. Subsequently, after three washes with PBS, the cells were then incubated with 1 µg/ml DAPI at room temperature for 1 h. The cells were analyzed using the Laser Scanning Confocal Microscope (OLYMPUS, FV1000).

## Lysate preparation, Click chemistry, Trypsin digestion, Peptide enrichment, and DTT/IAA elution

For cell surface lysine reactivity detection, the light cells were labeled by 50 µM OPA-S-S-alkyne and the heavy cells were labeled by 500 µM OPA-S-S-alkyne in the forward SILAC experiments. In the reverse SILAC experiments, the heavy cells were labeled by 50 µM OPA-S-S-alkyne and the light cells were labeled by 500 µM OPA-S-S-alkyne. After cell lysis and protein concentration determination, the light and heavy lysates (500 µg each) were combined and incubated in 500 µL DPBS buffer containing 100 µM biotin-PEG$_4$-azide, 2.5 mM freshly prepared sodium ascorbate, 500 µM BTTP, 250 µM CuSO$_4$ for 2 h at room temperature with rotation.

The click-labeled lysates were precipitated by 2 mL cold acetone at −20 °C overnight. The precipitated proteins were centrifuged at 15,000 × g for 30 min at 4 °C and washed twice with 1 mL cold methanol. Protein pellets were resuspended in 1 mL 50 mM ammonium bicarbonate and trypsin (enzyme: substrate ratio was 1:50) was then added. Trypsin digestion was performed at 37 °C with rotation overnight. Then 50 µL neutravidin beads (prewashed three times with PBS) were added to the solution, and the resulting mixture was incubated for 3 h at room temperature with rotation. The beads were pelleted by centrifugation and washed with 0.1% SDS in PBS three times, 8 M urea in PBS three times, 10 × PBS three times and distilled water three times to remove nonspecifically bound peptides. The modified peptides were released from the beads by two treatments of 100 µL 50 mM DTT at 37 °C for 30 min with rotation to cleave the disulfide bridge in the labeling reagents. The beads were then washed with 50% (v/v) ACN/H$_2$O containing 0.1% TFA, and the washes were combined with the eluent to form the cleavage fraction. The labeled peptides were desalted with C18 Zip-Tips (Merck) and then incubated with 100 mM iodoacetamide for 45 min at room temperature in the dark. The resulting peptides were desalted with C18 Zip-Tips and dried in a vacuum centrifuge.

## LC-MS/MS analysis

LC-MS/MS analysis of enriched peptides was performed on a nano-HPLC chromatography system connected to a hybrid trapped ion mobility spectrometry quadrupole time-of-flight mass spectrometer (TIMS-TOF Pro, Bruker Daltonics) via a CaptiveSpray nano-electrospray ion source. A total of 200 ng peptides dissolved in solvent A (0.1% formic acid) was loaded onto the analytical column (75 µm i.d. × 25 cm) and separated with a 60 min gradient (2 – 22% solvent B (ACN with 0.1% formic acid) for 45 min, 22 – 37% B for 5 min, 37 – 80% B for 5 min, and then 80% B for 5 min). The flow rate was maintained at 300 nL/min. For MS analysis, the accumulation and ramp time were set as 100 ms each. Survey full-scan MS spectra (m/z 100–1700) were obtained in positive electrospray mode. The ion mobility was scanned from 0.7 to 1.3 Vs/cm$^2$. The overall acquisition cycle of 1.16 s comprised one full TIMS-MS scan and 10 parallel accumulation-serial frag-mentation (PASEF) MS/MS scans. During PASEF MSMS scanning, the collision energy was ramped linearly as a function of the mobility from 59 eV at 1/K0 = 1.6 Vs/cm$^2$ to 20 eV at 1/K0 = 0.6 Vs/cm$^2$.

## Peptide identification and quantification

The raw MS data were searched against the Swiss-human database (downloaded on July 01, 2021, containing 20 381 protein sequence entries) using PEAKS ONLINE for peptide and protein identifications. The precursor mass error tolerance was set to 15 ppm, the fragment mass error tolerance was set to 0.05 Da, the maximum allowed missed cleavages of specific trypsin digestion was set to 5, and the length of the identified peptides was set to 6 – 45 amino acids. The fixed modifications were set to carbamidomethyl cysteine (+57.0215 Da), modifications on lysine (304.0882 and 312.1024 Da for light and heavy OPA-S-S-alkyne tagged adducts) and heavy arginine (10.0083 Da) were set as variable modifications. The false discovery rate (FDR) for peptide and protein group was set to less than 0.01. Quantification was also performed on PEAKS ONLINE by modified SILAC quantification method. The SILAC ratios of light and heavy MS1 peaks for each unique peptide were quantified.

## Assignment of cell surface lysines

The assignment of cell surface lysines was based on protein localization and its topological domain in Uniprot database. First, protein cellular localization was mapped by Uniprot database, and proteins with Gene Ontology cellular component (GO.CC) annotations of "plasma membrane", "cell surface", or "extracellular" (including the terms of "extracellular matrix", "extracellular region", "extracellular space", "extracellular vesicle", and "extracellular exosome") were termed as cell surface annotated proteins. And the site of probe-tagged lysines in assigned proteins was counted by the python program. The lysines from known transmembrane proteins (TMPs) were classified as located in extracellular, transmembrane, cytoplasmic region and no location information according to topological domain (extracellular/transmembrane/cytoplasmic) information of TMPs in Uniprot database. The extracellular lysines were definitely exposed on the cell surface. The lysines having no location information and lysines from other cell surface annotated proteins were potentially exposed on the cell surface. All lysine residues definitely and potentially exposed on the cell surface were included for the following reactivity analysis.

## R value calculation and processing

For reactivity measurements by SILAC-ABPP, overlapping peptides with the same modified lysine (for example, different charge states or tryptic termini) were grouped together and the median ratio was reported as the final ratio ($R$). A maximal ratio of 20 was assigned when peptides showed a ≥ 95% reduction in MS1 peak area. The peptide ratios reported by PEAKS ONLINE were further filtered to ensure the removal or correction of low-quality ratios in each individual data set. The quality filters applied were the following: removal of half tryptic peptides; for ratios with high standard deviations from the median (90% of the median or above) the lowest ratio was taken instead of the median; removing any peptides with low quality elution profiles (quality score <20). When aggregating data across experimental replicates, the mean of each experimental median $R$ was reported. The final reported ratio for a given lysine was the median ratio across the biological replicates (forward and reverse SILAC biological replicates).

## Protein class analysis

To place each transmembrane protein containing hyper- reactive lysines into a distinct protein class, these proteins were analyzed by PANTHER Protein Class tool. According to PANTHER Protein Class annotation, the proteins were classified as intercellular signal molecule (PC00207), membrane traffic protein (PC00150), protein modifying enzyme (PC00260), transfer/carrier protein (PC00219), transmembrane signal receptor (PC00197), cell adhesion molecule (PC00069), defense/immunity protein (PC00090) and others.

## Sequence motifs

For all lysines quantified in the reactivity profiling experiments, the flanking sequence (±8 amino acids) was determined with a custom python script, parsing the UniProtKB entries for all proteins identified. The sequences were binned by lysine reactivity (hyper-reactive: $R < 2$; moderately-reactive: $2 < R < 5$; low-reactive: $R > 5$) and evaluated for sequence motifs using WebLogo <http://weblogo.berkeley.edu/logo.cgi >.

## Recombinant expression of proteins by transient transfection

Recombinant proteins were produced by transient transfection of HEK293T cells with recombinant DNA. The following plasmid constructs (human proteins) were purchased from YouBia: wild-type plasmids pcDNA3.1-ROR2−3×FLAG and pcDNA3.1-ENG-3×FLAG, mutant plasmids pcDNA3.1-ROR2(K382R)-3×FLAG and pcDNA3.1-ENG(K285R)-3×FLAG. HEK 293T cells were grown to 50 % confluency in 5 mL DMEM supplemented with 10% FBS and 1% penicillin/streptomycin in 6 cm dishes. Three μg of DNA was diluted in 400 μL Opti-MEM and 6 μL of Lipofectamine™ 3000 (Invitrogen) were added. The mixture was incubated at room temperature for 20 min and added dropwise to the cells. Cells were grown for 48 h at 37 °C with 5% $CO_2$. Cells were harvested by scraping and lysed for western blot analysis of recombinant protein expression. And cells were labeled by 100 μM OPA-S-S-alkyne before harvesting cells for gel-based chemical proteomic assay.

## Western Blot analysis of recombinant protein expression

Cell lysates from WT and KR mutant cells with equal initial amounts of proteins were boiled for 10 min, and separated by SDS-PAGE, followed by transfer onto PVDF membranes. After blocking for 1 h with 5% milk, blots were incubated overnight with primary antibodies (1:1000 anti-FLAG, Cell Signaling Technology or 1:100000 anti-ACTIN, ABclonal) at 4 °C. Blots were then washed (3 × 5 min, TBS-T) and incubated with peroxidase-conjugated secondary antibodies with a 1:1000 dilution for 1 h at room temperature. Blots were further washed (3 × 5 min, TBST) and visualized using ECL (GE Healthcare) scanned by ImageQuant ECL Imager (GE Healthcare Life Sciences).

## Assessment of lysine hyper-reactivity of recombinant proteins by gel-based ABPP

Cell lysates from OPA-S-S-alkyne labeled WT and KR mutant cells with equal initial amounts of proteins were reacted with azide-PEG$_4$-biotin respectively. The resulting biotinylated proteins were enriched by neutravidin beads and eluted by heating beads at 95 °C in loading buffer containing 10% SDS for 10 min. The enriched proteins were separated by SDS-PAGE and analyzed by western blot using flag antibody as described above.

## In vivo treatment of OPA-S-S-alkyne and interaction assay of ENG and BMP9 by western blot

Hela cells were grown to 50 % confluency in DMEM supplemented with 10% FBS and 1% penicillin/streptomycin. The plasmid constructs used (human proteins) were same as above: wild-type plasmids pcDNA3.1-ROR2-3×FLAG and pcDNA3.1-ENG-3×FLAG. All plasmids were transfected to cells by polyethylenimine (Polysciences) at 37 °C for 48 h. Then cells were seeded onto 6-well plates, grown for 48 h at 37 °C with 5% CO2 and starved overnight to avoid the interference of BMP9 in serum. Then Hela cells were subjected to three washes with room-temperature PBS (pH 7.4), followed by the addition of room-temperature PBS containing OPA-S-S-alkyne with various concentrations. The labeling reaction was gently agitated on an orbital shaker at room temperature for 10 min and then quenched using 1 M glycine. The labeled cells were washed three times with PBS and incubated with 25 ng/ml BMP9 at 4 °C for 2 h. After incubation, cells were washed three times with PBS and then lysed for western blot. Cell lysates from WT and KR mutant cells with equal initial amounts of proteins were boiled for 10 min, and separated by SDS-PAGE, followed by transfer onto NC membranes. After blocking for 15 min with rapid blocking buffer, blots were incubated overnight with primary antibodies (1:2000 anti-BMP9-FITC, Cloud-Clone Corp or 1:5000 anti-Tubulin, ABclonal) at 4 °C. Blots were then washed (3 × 5 min, TBST) and incubated with secondary antibodies with a 1:5000 dilution for 1 h at room temperature. Blots were further washed (3 × 5 min, TBST) and visualized using ECL (GE Healthcare).

## Protein domain analysis

For all lysines quantified in transmembrane proteins, Uniprot was used to obtain the domain information of corresponding information. Python was used for data processing.

## PTM analysis

The PTM analysis was based on the 2022 realized protein lysine modifications database, CPLM 4.0 and Python was used for data processing.

## Reporting summary

Further information on research design is available in the Nature Portfolio Reporting Summary linked to this article.

## Data availability

The source data underlying Figs. 1c, 1e, 2b, 2d, 4b, and Supplementary Figs. 8, 22, 33 are provided as a Source Data file. The proteomics data have been deposited to the ProteomeXchange Consortium via iProx with the dataset identifier PXD042888. All other data are available from the corresponding authors upon reasonable request. Source data are provided with this paper.

## Code availability

The Python code used for proteomic data processing are deposited to Github repository at "https://github.com/FudanLuLab/useful-utilities/blob/main/protein_data_extractor.py".

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

## Acknowledgements

This work is supported by the National Key Research and Development Program of China (2021YFA1301602 to Y.Z.), the National Natural Science Foundation of China (Grants 22174021 to Y. Z. and 82121004 to H. L), Shanghai Project (22142202400 to H.L.), and the innovative research team of high-level local university in Shanghai (to Y.Z. and H.L.).

## Author contributions

Y.Z., T.W., and S.M. conceived and designed the study; T.W. and S.M. conducted all of the experiments; T.W., S.M., G.J., G.W., Y.L., and L.Z. performed data analysis and Y.Z., T.W., and S.M. wrote the manuscript. Y.Z. and H.L. contributed validation, supervision and funding acquisition. All authors have given approval to the final version of the manuscript.

## Competing interests

The authors declare no competing interests.
