## [Peer Review File · Nature Communications]

Reviewers' Comments:

Reviewer #1:

Remarks to the Author:

In their study, Wang et al. develop a method to study the reactivity of lysine residues specifically on the cell surface. For this purpose, they develop a probe based on an ortho-phthalaldehyde reactive group to react with lysines. The probe furthermore contains an alkyne for enrichment and a disulfide that will get cleaved in cells in order to increase selectivity for cell membrane proteins. They use this probe first on isolated peptides and proteins and then in living cells, where they study the reactivity by fluorescence labeling and by mass spectrometry. They show high selectivity for the labeling of extracellular proteins and identify highly reactive lysines in several cell membrane proteins, which was previously not possible.

The study addresses the highly contemporary field of developing covalent inhibitors for other amino acids than cysteines. In this context, lysine is surely a very interesting alternative and the angle to selectively label cell-surface lysines is highly interesting. The data convincingly underlines the claims to a large extent. I am, therefore, convinced that this manuscript could be suitable for publication in Nature Communications, after some points are addressed.

1. The conclusion of the paper is very short. The authors should discuss the implications of their findings a bit more thoroughly. Especially, it is very interesting that the probe shows such high selectivity for extracellular labeling, but there is no real explanation given. The structure of the probe, apart from the contribution of the cleavable linker, does not intuitively explain this observation. The authors should attempt to give an explanation for this observation.
2. The fluorophore the authors use to monitor the selective labeling of cell surface proteins is most likely itself not cell permeable as it contains sulfonate groups. The authors should perform a control with another probe that leads to intracellular labeling. In that way, they can show that, if intracellular labeling occurs, this can also be monitored in their setup to exclude that the selectivity originates from the dye and not the probe. If the selectivity is based on the fluorophore, this experiment needs to be repeated in a different setup that allows studying the true selectivity of the probe.
3. The authors claim in the abstract and main body of the text that cell surface proteins are labeled with >90% selectivity. Nevertheless, only about 80% of the proteins in the proteomics are cell membrane proteins (Figure 2e). The value of >90% originates from previous filtering for cell membrane proteins, which is not really a fair thing to do. This statement should be adjusted to >80% in the abstract and main text.
4. The comparison to the study of Hacker et al. is not entirely conclusive. While it is true that this study did not identify many cell membrane proteins, it was also performed in soluble lysate of cells and, therefore, the presence of cell membrane proteins is not really expected. This should be mentioned, as it is to the best of my knowledge not established, how the labeling of STP-alkyne would look like in the context of a living cell.
5. On page 9: "The probe concentration of 10 μ M is enough for the protein labeling, and this labeling concentration is almost one order of magnitude lower than that of traditional NHS labeling." As there is no direct comparison to NHS esters given in the manuscript, I believe

that this statement has to be deleted.

6. The blots in Figure 4b need to be replicated and quantified.

7. All proteomics data must be deposited to an open repository such as EMBL PRIDE.

8. The entire manuscript needs to be carefully proofread for grammatical and textual mistakes.

Reviewer #2:

Remarks to the Author:

Reviewer's overall assessment:

Wang and co-workers present a chemical proteomic methodology GASF to globally map the reactivity and functionality of cell surface lysine in living cells. Authors report >90% specificity towards cell surface lysines. Authors designed a new chemical proteomic probe OPA-S-S-alkyne that can efficiently and selectively target the lysine exposed on the cell surface. Authors claim that selectivity is due to cleavage of disulfide bond in the linker of OPA-S-S-alkyne probe induced by reducing cytoplasmic environment. Authors quantified 2639 cell surface lysines with >90% specificity in Hela cell and several hundred residues with heightened reactivity. Authors showed that hyper-reactive lysine residues K382 on ROR2 and K285 on ENG are at the protein interaction interface, which was initially based on co-crystal structures of protein complexes. Authors further claim that their chemical proteomic strategy is general and can be applicable to discover a variety of ligandable amino acid like cysteine on cell surface by modifying the warhead of the chemical proteomic probe with different functional group towards these amino acids.

The article is timely but provides an insufficient addition to the existing literature. The scope of the article is clear, but the ideas are inadequately presented and discussed. The Supporting Information is not of good standard and lacks essential materials. Overall, this report is fairly insignificant and will not attract broad range of readership of Nature Communication in its original form. Therefore, this reviewer does not recommend publication of this manuscript to Nature Communication.

Reviewer's questions, comments, and recommended modifications to the manuscript:

1) Authors' claim that "Among the small molecule drugs approved by FDA, drugs targeting membrane proteins account for more than 60% of all clinical drugs" is not supported by the reference paper number three.

2) Authors claim that "new chemical proteomic probe OPA-S-S-alkyne that can efficiently and selectively target the lysine exposed on the cell surface and develop a chemical proteomics strategy for global analysis of surface functionality (GASF) in living cells". Authors rely on the presence of reducing cytoplasmic environment, which is directly related to intracellular concentrations of GSH (glutathione), to cleave the disulfide linker and reduce

the labeling of intracellular proteins. Because the intracellular concentrations of GSH vary drastically between different cell lines (due to different levels of oxidative stress), this strategy may display selectivity towards surface protein lysines only when intracellular GSH concentrations are high enough to sufficiently cleave the disulfide linker of the probe.

3) Quantification and statistical analysis of western blots in Figure 4 are missing. This will be crucial for the conclusion drawn by authors about ROR2 and ENG proteins.

4) Authors should have determined the effect of intracellular GSH concentration on efficiency of disulfide cleavage in the linker.

5) Authors should have quantified lysines on membrane-bound proteins labeled by probe that are exposed to extracellular matrix.

6) Authors should have separated soluble and particulate proteomes to further provide evidence of probe labeling selectivity of membrane-bound proteins in Figure 2.

Reviewer's recommended modifications to the Supporting Information:

1) Images of ¹H- and ¹³C-NMR spectra for all new compounds and intermediates are missing from the Supporting Information and must be included at the end of the document. This is a standard practice.

2) Full images of uncropped and unprocessed gels and western blots are missing from the Supporting Information and must be included in the Source Data. This is a standard practice for all Nature journals.

3) All raw proteomics data must be uploaded to the PRIDE or other repositories.

4) All custom python code used for proteomic data processing must be deposited to Github repository.

Reviewer #3:

Remarks to the Author:

Summary: In this manuscript the authors develop a new labeling reagent and protocol for identifying and analyzing cell surface proteins with reactive lysine residues. The reagent relies on ortho-phthaldehyde reactivity with amine groups linked via disulfide to an alkyne moiety for detection (via click chemistry). This approach expands the known collection of reactive lysine residues on the cell surface, which is a valuable trove of information for scientists interested in developing probes or drugs to covalently target cell surface proteins. The primary thrust of this paper (proteomics analysis of labeling of cellular proteins) is clear and seems to be well done. However, further context needs to be provided for the history of

Lys-targeting probes. Additional details and experiments must be run to characterize the effects of the labeling probe on cells. The paper would be more impactful if it could provide Lysine labeling information from an additional cell type or condition.

Overall, this is a valuable contribution to the field that requires some improvements before acceptance.

Major comments:

*Selective labeling of cell surface proteins (with biotin) has been achieved using amine reactive N-hydroxysuccinimide ester chemistry for over 30 years (ref: <https://doi.org/10.1007/BF01871942>). Use of a disulfide linkage to ensure restriction of labeling to cell surface proteins (biotin-ss-NHS) has been used for over 25 years (<https://doi.org/10.1152/ajprenal.1995.268.2.F285>). Some historical context should be provided in the text to emphasize the ways in which the approach described in this paper differs from more commonly used methods. The goals of this work differ from past work but it's not clear how the chemical labeling methods used in this manuscript compare to more common methods like NHS chemistry. Is there a reason to use OPA instead of NHS for amine labeling? In that vein, can the authors comment on whether pre-treatment with an NHS ester would block labeling with the OPA-SS-alkyne probe?

*The assay used to measure cell viability (Figure 2d), covers a very short time frame. Although cells are only exposed to labeling agent for a short duration it is possible that they are exerting toxic effects on cells that will not be obvious in cell viability assays after only 10 minutes of measurement. The claim that the labeling technique is non-toxic would be strengthened by alternative assays that measure membrane integrity, cell viability over longer durations, or membrane trafficking assays (transferrin receptor internalization).

*Is OPA-SS-alkyne cell permeable? If the disulfide linkage of OPA-SS-alkyne is reduced under cytoplasmic conditions, this means that ortho-phthalaldehyde might still enter the cell and react with intracellular lysines, potentially confounding biological effects attributed to cell surface labeling. The authors should address this question. One possibility is to create a "non-reducible" version ("OPA-alkyne") and see how the complement of proteins labeled with this probe compare to those labeled OPA-SS-alkyne

*On page 7 the text says, "The confocal fluorescence images of K285R mutant also showed diminished colocalization between BMP9 and ENG (Figure S17), which proved that K285 also has an obvious influence on the colocalization of BMP9 and ENG." Fluorescence colocalization is an imprecise and somewhat unreliable method for measuring interactions of cell surface proteins. This is particularly true after fixing cells and only showing one data set with a few cells per field of view. Another confusing aspect of these images is that it appears that BMP9 is mostly cytoplasmic in WT ENG conditions (but not other conditions). The claim that BMP9-ENG interactions are disrupted upon labeling is not convincing and the authors should consider other methods to strengthen these claims. Also, the phrase "proved" should be avoided in scientific writing.

*To demonstrate the generalizability of this platform, more than one cell type/condition should be evaluated. Since the authors already perform experiments in HEK293 cells it could be interesting to compare GASF labeling in HEK293 to labeling in HeLa. This would provide valuable insights into the extent to which protein labeling varies between different cell lines.

Minor comments:

*The authors design their probe (OPA-SS-alkyne) to preferentially label cell surface proteins by incorporation of a disulfide linkage that is hypothesized to be reduced under cytoplasmic conditions. This design should be evaluated further by demonstrating that this disulfide is in fact reduced under conditions that mimic the cytoplasm.

*The authors should comment on whether OPA-SS-alkyne reacts with protein N-termini. Is this seen in proteomics experiments? If this is unclear, it can be tested with a peptide with a free terminus without a Lysine sidechain

*On page 7, the paragraph starting "Except common drug target and ligand..." doesn't make sense. The authors should revise this sentence.

*In experiments in which HEK293T cells are transfected with ENG and BMP9 the authors should comment on whether HEK293T cells endogenously express one or both proteins. This factor could influence the findings in this cell line.

*On page 7 the text says "nearly half (108/224) of the highly reactive lysines were annotated to be PTM sites..." How can these lysines be labeled if they are already modified with PTMs? Further explanation is needed

*Complementary enzymatic methods to measure protein N-termini exposed on the cell surface should be mentioned and discussed (<https://www.pnas.org/doi/10.1073/pnas.2018809118>)

*Affinity-based methods for labeling specific N-terminal amines (<https://onlinelibrary.wiley.com/doi/full/10.1002/anie.201403214>) or lysine residues (<https://pubs.acs.org/doi/full/10.1021/acs.bioconjchem.2c00334>) should be discussed for context

*The word "obvious" is used frequently throughout the text. Many of the assertions did not seem obvious to me.

Reviewer #1

1. In their study, Wang et al. develop a method to study the reactivity of lysine residues specifically on the cell surface. For this purpose, they develop a probe based on an ortho-phthalaldehyde reactive group to react with lysines. The probe furthermore contains an alkyne for enrichment and a disulfide that will get cleaved in cells in order to increase selectivity for cell membrane proteins. They use this probe first on isolated peptides and proteins and then in living cells, where they study the reactivity by fluorescence labeling and by mass spectrometry. They show high selectivity for the labeling of extracellular proteins and identify highly reactive lysines in several cell membrane proteins, which was previously not possible. The study addresses the highly contemporary field of developing covalent inhibitors for other amino acids than cysteines. In this context, lysine is surely a very interesting alternative and the angle to selectively label cell-surface lysines is highly interesting. The data convincingly underlines the claims to a large extent. I am, therefore, convinced that this manuscript could be suitable for publication in Nature Communications, after some points are addressed.

Response: We appreciate the reviewer's positive feedback. Detailed response to each comment is listed below.

2. The conclusion of the paper is very short. The authors should discuss the implications of their findings a bit more thoroughly. Especially, it is very interesting that the probe shows such high selectivity for extracellular labeling, but there is no real explanation given. The structure of the probe, apart from the contribution of the cleavable linker, does not intuitively explain this observation. The authors should attempt to give an explanation for this observation.

Response: We thank the reviewer for this suggestion. As far as we know, the most commonly used probe for lysine labeling is N-Hydroxysuccinimide (NHS). Compared with NHS, OPA shows less side reactions, more efficiency of reaction with lysines and more stability of adducts.¹ Importantly, the OPA can rapidly and efficiently label lysine residues very fast (within 1 minute as shown in Figure 1c, 1e and 2b). As a result, we

chose OPA as warhead in this work to reduce the intracellular labeling as much as possible and avoid side reactions. Plus, considering the specificity of cell surface labeling, we designed a disulfide bond in our probe.

Therefore, considering the questions mentioned by other reviewers, description of the probe design ideas is modified in the manuscript: “Central to this probe is 1) an electrophilic warhead for covalent binding of lysine on proteins. The covalent reaction used in this study was chosen based on ortho-phthalaldehyde (OPA) and amine conjugation to enable the highly efficient and selective lysine labeling because OPA can react chemoselectively with lysine rapidly under the physiological condition via formation of phthalimidines. To be noted, the ultra-fast reaction between OPA and lysine may reduce the intracellular labeling as much as possible; 2) a linker that tunes the reaction as well as minimizes undesirable labeling on inner cell proteins. Herein, a disulfide-linker is designed in this probe as it is known that the disulfide-linker helps enhance the purity of following cell surface protein fraction. 3) an affinity tag for purification of labeled membrane proteins from their native environment. For this purpose, we installed an alkyne into the probe and then used the click chemistry between alkyne-functionalized probe and azido-biotin assisted by CuAAC considerations to introduce a biotin tag on the labeled proteins for further streptavidin purification.”

In addition, we added some experiments. We compared the properties of cells labeled with OPA-alkyne and OPA-S-S-alkyne, finding that OPA-S-S-alkyne had better selectivity for cell surface proteins than OPA-alkyne (Figure R1 and R2). We also explored the ability of GSH to reduce S-S and found that it was able to reduce part of the disulfide bond in a short time (Figure R4). So even if a small amount of probe entered the cell, it would be rapidly reduced and thus not enriched. Therefore, we assume that the high selectivity of OPA-S-S-alkyne can be attributed to a combination of factors, including low cell permeability, short labeling time, and the reduction of disulfide bonds.

We also modified the conclusion parts correspondingly.

3. The fluorophore the authors use to monitor the selective labeling of cell surface proteins is most likely itself not cell permeable as it contains sulfonate groups. The authors should perform a control with another probe that leads to intracellular labeling. In that way, they can show that, if intracellular labeling occurs, this can also be monitored in their setup to exclude that the selectivity originates from the dye and not the probe. If the selectivity is based on the fluorophore, this experiment needs to be repeated in a different setup that allows studying the true selectivity of the probe.

Response: We thank the reviewer for this suggestion. As mentioned by the reviewer, we made another probe, OPA-alkyne, which is cell permeable to monitor the selective labeling of cell surface proteins. To avoid the issue that the fluorophore is not cell permeable, we monitored the selective labeling of cell surface proteins by following experiment setup. After the OPA-S-S-alkyne or OPA-alkyne procedure, the cells were fixed with cold methanol at 4 °C for 15 min. Then the cells were washed three times with PBS followed by the permeabilization with 0.2% Triton X-100 in PBS for 10 min at room temperature. The cells were washed three times with PBS, followed by incubation for 2 h in PBS containing 100 µM TAMRA Azide, 2.5 mM sodium ascorbate, 500 µM BTTP, 250 µM CuSO₄ at room temperature in the dark. Subsequently, after three-time washes with PBS, the cells were then incubated with 1 µg/ml DAPI (4',6-diamidino-2-phenylindole) at room temperature for 1 h. The cells were analyzed using the Laser Scanning Confocal Microscope. As shown in Figure R1, OPA-S-S-alkyne selectively labeled cell surface proteins, whereas OPA-alkyne labeled both cell surface and intracellular proteins.

Figure R1. Cells were labeled by OPA-S-S-alkyne (left) and OPA-alkyne (right), followed by fixation, permeabilization, click with TAMRA Azide and analyzed by confocal fluorescent microscopy.

4. The authors claim in the abstract and main body of the text that cell surface proteins are labeled with >90% selectivity. Nevertheless, only about 80% of the proteins in the proteomics are cell membrane proteins (Figure 2e). The value of >90% originates from previous filtering for cell membrane proteins, which is not really a fair thing to do. This statement should be adjusted to >80% in the abstract and main text.

Response: We sincerely apologize for the lack of clarity thank the reviewer for pointing that out. The corrected statement was updated in the revised manuscript.

5. The comparison to the study of Hacker et al. is not entirely conclusive. While it is true that this study did not identify many cell membrane proteins, it was also performed in soluble lysate of cells and, therefore, the presence of cell membrane proteins is not really expected. This should be mentioned, as it is to the best of my knowledge not established, how the labeling of STP-alkyne would look like in the context of a living cell.

Response: We thank the reviewer for pointing that out. As mentioned by the reviewer, the labeling of STP-alkyne in living cell is not established. The study of Hacker et al, though not in living cells, but is one of the most comprehensive datasets among the current researches on ligandable lysines. They applied STP-alkyne to label lysines in cell lysates and they fractionated cell pellets to yield soluble and membrane fractions and then both fractions were labeled by STP-alkyne. Based on these, we compared our data with Hacker' data. Comparison of these two datasets showed that our method can probe more cell surface lysines than that using traditional cell fractions fractionation by ultracentrifugation. The corrected statement was updated in the revised manuscript.

6. On page 9: "The probe concentration of 10 μ M is enough for the protein labeling,

and this labeling concentration is almost one order of magnitude lower than that of traditional NHS labeling.” As there is no direct comparison to NHS esters given in the manuscript, I believe that this statement has to be deleted.

Response: We thank the reviewer for this suggestion. This data is based on the reference.¹ Such statement was deleted in the revised manuscript.

7. The blots in Figure 4b need to be replicated and quantified.

Response: We thank the reviewer for this suggestion. We have replicated the experiments in Figure 4b and statistical analysis of western blots in Figure 4b were carried out(Figure R2).

Figure R2. Both ROR2 and ENG were labeled less by OPA-S-S-alkyne when the hyper-reactive lysine were mutated to arginine.

8. All proteomics data must be deposited to an open repository such as EMBL PRIDE.

Response: We thank the reviewer for this suggestion. All proteomics data have been deposited to iProx.

URL: <https://www.iprox.cn/page/PSV023.html?url=1701915523347Dtf>

Passport: y4xB

9. The entire manuscript needs to be carefully proofread for grammatical and textual mistakes.

Response: We thank the reviewer for this suggestion. We have carefully proofread for grammatical and textual mistakes and updated in the revised manuscript.

Reviewer #2

Wang and co-workers present a chemical proteomic methodology GASF to globally map the reactivity and functionality of cell surface lysine in living cells. Authors report >90% specificity towards cell surface lysines. Authors designed a new chemical proteomic probe OPA-S-S-alkyne that can efficiently and selectively target the lysine exposed on the cell surface. Authors claim that selectivity is due to cleavage of disulfide bond in the linker of OPA-S-S-alkyne probe induced by reducing cytoplasmic environment. Authors quantified 2639 cell surface lysines with >90% specificity in HeLa cell and several hundred residues with heightened reactivity. Authors showed that hyper-reactive lysine residues K382 on ROR2 and K285 on ENG are at the protein interaction interface, which was initially based on co-crystal structures of protein complexes. Authors further claim that their chemical proteomic strategy is general and can be applicable to discover a variety of ligandable amino acid like cysteine on cell surface by modifying the warhead of the chemical proteomic probe with different functional group towards these amino acids. The article is timely but provides an insufficient addition to the existing literature. The scope of the article is clear, but the ideas are inadequately presented and discussed. The Supporting Information is not of good standard and lacks essential materials. Overall, this report is fairly insignificant and will not attract broad range of readership of Nature Communication in its original form. Therefore, this reviewer does not recommend publication of this manuscript to Nature Communication.

Reviewer's questions, comments, and recommended modifications to the manuscript:

- 1.** Authors' claim that "Among the small molecule drugs approved by FDA, drugs targeting membrane proteins account for more than 60% of all clinical drugs" is not supported by the reference paper number three.

Response: We sincerely apologize for the mistake. We have changed the reference as “The in silico human surfaceome” PNAS, 2018, 115 (46) E10988-E10997 <https://doi.org/10.1073/pnas.1808790115>”.

2. Authors claim that “new chemical proteomic probe OPA-S-S-alkyne that can efficiently and selectively target the lysine exposed on the cell surface and develop a chemical proteomics strategy for global analysis of surface functionality (GASF) in living cells”. Authors rely on the presence of reducing cytoplasmic environment, which is directly related to intracellular concentrations of GSH (glutathione), to cleave the disulfide linker and reduce the labeling of intracellular proteins. Because the intracellular concentrations of GSH vary drastically between different cell lines (due to different levels of oxidative stress), this strategy may display selectivity towards surface protein lysines only when intracellular GSH concentrations are high enough to sufficiently cleave the disulfide linker of the probe.

Response: We thank the reviewer’s comments. Use of a disulfide linkage to improve the specificity of labeling to cell surface proteins (biotin-S-S-NHS) has been reported in previous study.² Therefore, when we designed the probe, we borrowed this concept and tried to minimize the labeling of intracellular proteins. To do so, we also labeled the living cell with OPA-alkyne probe which did not have the S-S linker and characterized the OPA-alkyne probe labeled proteome also. The comparison of these two probes showed that labeling with the OPA-S-S-alkyne obtained a much higher selectivity of cell-surface proteome (Figure R1 and R3). We therefore also chose OPA-S-S-alkyne according to the comparison results of the two probes. We assumed that the high selectivity of OPA-S-S-alkyne is attributed to a combination of factors, including

low cell permeability of the probe, short labeling time, and the reduction of disulfide bonds. Here we are sorry for your misunderstanding.

Figure R3. GOCC analysis of OPA-alkyne labeled proteins (top) and OPA-S-S-alkyne labeled proteins (bottom) from HeLa cells.

3. Quantification and statistical analysis of western blots in Figure 4 are missing. This will be crucial for the conclusion drawn by authors about ROR2 and ENG proteins.

Response: We thank for the reminder. Quantification and statistical analysis of western blots in Figure 4 are provided in this revision (Figure R2).

4. Authors should have determined the effect of intracellular GSH concentration on efficiency of disulfide cleavage in the linker.

Response: We thank the reviewer's comments. We investigated the concentration of

GSH on efficiency of disulfide cleavage in the linker. OPA-S-S-alkyne labeled BSA was mixed with different concentrations of GSH for 10 min and quenched by chloroform-methanol precipitation. Then protein pellets were resuspended in 0.4% SDS in PBS and clicked with TAMRA Azide. Proteins were visualized by SDS-PAGE and in-gel fluorescence scanning as described above. As shown in Figure R4, the fluorescence intensity decreased with the increasing concentration of GSH, indicating the GSH could partly cleave the disulfide. Therefore, we reasoned that even if the probe enters the cell, the protein in the cell will not be captured during the following purification step. But we cannot rule out disulfide as the only cause. In combination with the first question raised by the reviewer 1, we modified the description in the manuscript.

Figure R4. OPA-S-S-alkyne labeled BSA was mixed with different concentrations of GSH, showing concentration-dependent decrease in fluorescence intensity. The gels were stained by Coomassie brilliant blue to show equal loading.

5. Authors should have quantified lysines on membrane-bound proteins labeled by probe that are exposed to extracellular matrix.

Response: We thank the reviewer's comments. According to our data, in total, 2687 lysine residues assigned to 831 cell surface annotated proteins were quantified across 6 replicate experiments. Among these 831 proteins, 216 known TMPs (transmembrane

proteins) have topological domain information in Uniprot database. From the 216 TMPs, 718 lysines were quantified, of which 656 lysines corresponded to 197 TMPs were located in the extracellular region (Table S2). The remaining 48 lysines were located in the cytoplasmic or transmembrane region and 14 lysines had no location information (Figure 3b). Therefore, the proportion of extracellular lysines we identified was about 91%.

6. Authors should have separated soluble and particulate proteomes to further provide evidence of probe labeling selectivity of membrane-bound proteins in Figure 2.

Response: We thank the reviewer's comments. We labeled the HeLa cells by the probe and then separated soluble and particulate proteomes. These two parts and total proteins are blotted (Figure R5), showing that OPA-S-S-alkyne mainly labeled membrane proteins.

Figure R5. Non-fractionated proteins (whole cell lysate), proteins from membrane and cytosolic fractions extracted from OPA-S-S-alkyne labeled HeLa cells were clicked with TAMRA Azide and analyzed by SDS-PAGE and in-gel fluorescence scanning (right).

Gels were colored by Coomassie Brilliant Blue (left) to confirm presence of all proteins in the fractions.

Reviewer's recommended modifications to the Supporting Information:

1) Images of ¹H- and ¹³C-NMR spectra for all new compounds and intermediates are missing from the Supporting Information and must be included at the end of the document. This is a standard practice.

Response: We apologize we could not provide the ¹H- and ¹³C-NMR spectra for all intermediates. The probe OPA-S-S-alkyne is designed by ourselves and synthesized by Serinno holdings limited. As a custom-made compound, the final product OPA-S-S-alkyne is now commercially available from Serinno holdings limited and the characterization of this probe is provided in Figure S1, S2, S3 and S4.

2) Full images of uncropped and unprocessed gels and western blots are missing from the Supporting Information and must be included in the Source Data. This is a standard practice for all Nature journals.

Response: Thanks for the reminder. Full images of uncropped and unprocessed gels and western blots are provided in source data.

3) All raw proteomics data must be uploaded to the PRIDE or other repositories.

Response: We thank the reviewer for this suggestion. All proteomics data have been deposited to iProx.

URL: <https://www.iprox.cn/page/PSV023.html?url=1701915523347DtfD>

Passport: y4xB

4) All custom python code used for proteomic data processing must be deposited to Github repository.

Response: Thank you for the reminder. The Python code used for proteomic data processing are deposited to Github repository at "https://github.com/FudanLuLab/useful-utilities/blob/main/protein_data_extractor.py".

Reviewer #3

Summary: In this manuscript the authors develop a new labeling reagent and protocol for identifying and analyzing cell surface proteins with reactive lysine residues. The reagent relies on ortho-phthaldehyde reactivity with amine groups linked via disulfide to an alkyne moiety for detection (via click chemistry). This approach expands the known collection of reactive lysine residues on the cell surface, which is a valuable trove of information for scientists interested in developing probes or drugs to covalently target cell surface proteins. The primary thrust of this paper (proteomics analysis of labeling of cellular proteins) is clear and seems to be well done. However, further context needs to be provided for the history of Lys-targeting probes. Additional details and experiments must be run to characterize the effects of the labeling probe on cells. The paper would be more impactful if it could provide Lysine labeling information from an additional cell type or condition. Overall, this is a valuable contribution to the field that requires some improvements before acceptance.

Response: We are very encouraged by the reviewer's comments. Response to each comment is listed below.

Major comments:

1. Selective labeling of cell surface proteins (with biotin) has been achieved using amine reactive N-hydroxysuccinimide ester chemistry for over 30 years (ref: <https://doi.org/10.1007/BF01871942>). Use of a disulfide linkage to ensure restriction of labeling to cell surface proteins (biotin-ss-NHS) has been used for over 25 years (<https://doi.org/10.1152/ajprenal.1995.268.2.F285>). Some historical context should be provided in the text to emphasize the ways in which the approach described in this paper differs from more commonly used methods. The goals of this work differ from past work but it's not clear how the chemical labeling methods used in this manuscript compare to more common methods like NHS chemistry. Is there a reason to use OPA instead of NHS for amine labeling? In that vein, can the authors comment on whether

pre-treatment with an NHS ester would block labeling with the OPA-SS-alkyne probe?

Response: We sincerely appreciate the reviewer for the suggestion. Relevant content has been added to the introduction. Using NHS as a warhead may also identified a number of different sites. However, it also shows reactivity with several amino acids other than lysine³. Most importantly, in this work we expect specific labeling of membrane proteins, so it is critical to use a quick labeling reaction and avoid intracellular reactions. So, considering that OPA shows less side reactions, more efficiency of reaction with lysines and more stability of adducts compared with NHS¹ as well as high reaction efficiency, we chose OPA as a warhead. Due to the amino reactivity of NHS, we infer that pretreatment with an NHS ester may partly block the OPA-SS-alkyne label. In addition, there are several amino warheads, and we believe that more sites may be found by using different warheads

2. The assay used to measure cell viability (Figure 2d), covers a very short time frame. Although cells are only exposed to labeling agent for a short duration it is possible that they are exerting toxic effects on cells that will not be obvious in cell viability assays after only 10 minutes of measurement. The claim that the labeling technique is non-toxic would be strengthened by alternative assays that measure membrane integrity, cell viability over longer durations, or membrane trafficking assays (transferrin receptor internalization).

Response: We are sorry for the inaccurate statement. The processing time of 10 minutes is based on the time for efficient labeling and the label time in the following SILAC-ABPP experiments. We agree that the claim of non-toxic labeling technique is inaccurate. Related statement has been revised to “The cell viability has no significant change after 10 min treatment of the probe”.

3. Is OPA-SS-alkyne cell permeable? If the disulfide linkage of OPA-SS-alkyne is reduced under cytoplasmic conditions, this means that ortho-phthaldahyde might still enter the cell and react with intracellular lysines, potentially confounding biological effects attributed to cell surface labeling. The authors should address this question. One

possibility is to create a “non-reducible” version (“OPA-alkyne”) and see how the complement of proteins labeled with this probe compare to those labeled OPA-SS-alkyne

Response: We sincerely appreciate the reviewer for the suggestion. We labeled the living cell with OPA-alkyne probe. And the comparison of both Fluorescence confocal image (Figure R1) and GOCC based on LC/MS-MS results (Figure R3) towards these two probes showed that labeling with OPA-S-S-alkyne obtained a much higher selectivity of cell-surface proteome. So we thought that the high selectivity of OPA-S-S-alkyne can be attributed to a combination of factors, including low cell permeability, short labeling time, and the reduction of disulfide bonds.

4. On page 7 the text says, “The confocal fluorescence images of K285R mutant also showed diminished colocalization between BMP9 and ENG (Figure S17), which proved that K285 also has an obvious influence on the colocalization of BMP9 and ENG.” Fluorescence colocalization is an imprecise and somewhat unreliable method for measuring interactions of cell surface proteins. This is particularly true after fixing cells and only showing one data set with a few cells per field of view. Another confusing aspect of these images is that it appears that BMP9 is mostly cytoplasmic in WT ENG conditions (but not other conditions). The claim that BMP9-ENG interactions are disrupted upon labeling is not convincing and the authors should consider other methods to strengthen these claims. Also, the phrase “proved” should be avoided in scientific writing.

Response: We are really grateful to the reviewer for the suggestion. We revised the manuscript accordingly. Regarding the phenomenon of BMP9 is mostly cytoplasmic in WT ENG conditions, we hypothesize that this is due to the interaction between BMP9 and WT ENG, forming a complex that enters the cell, while the interaction between BMP9 and KR ENG is relatively weak. As BMP9 is a secreted protein present in serum, approximately 10 ng/ml,⁷ we did not supplement additional BMP9 or induce serum starvation in our immunofluorescence experiments. This was done to observe the differences in BMP9 and ENG under untreated conditions in both cell types. Therefore,

the cytoplasmic BMP9 detected in these experiments may originate from endogenous or serum sources. In the WT condition, it still interacts with ENG, leading to the detection of a protein complex in the cytoplasm. We also quantified the BMP9 without the label of OPA-S-S-alkyne (Figure S16 and R6), and observed that the amount of BMP9 were lower in the K285R mutant compared to the wild type (WT), which is consistent with the immunofluorescence.

Figure R6. The intensity of BMP9 in K285-ENG was lower than the WT-ENG without the treatment of OPA-S-S-alkyne.

5. To demonstrate the generalizability of this platform, more than one cell type/condition should be evaluated. Since the authors already perform experiments in HEK293 cells it could be interesting to compare GASF labeling in HEK293 to labeling in HeLa. This would provide valuable insights into the extent to which protein labeling varies between different cell lines.

Response: Thank you for the suggestion. We labeled HEK293T cells with 100 μ M OPA-S-S-alkyne and applied the same procedure for enrichment and LC-MS/MS detection. We observed similar selectivity of our probe for labeling HEK293T cells, with 81% of the labeled lysines belonging to cell surface proteins (Figure R6a).

Additionally, GOCC analysis of the labeled proteins in HEK293T cells showed similar results to those in Hela cells (Figure R6b). This indicates that the GASF is also applicable to the HEK293T cell line.

Figure R6. a) Living HEK293T cells were labeled with OPA-S-S-alkyne (100 μM) followed by protein extraction, reacted with azide-biotin, digested by trypsin, enriched by avidin beads, eluted by DTT/IAA and analyzed by LC-MS/MS. A total of 1851 probe-labeled lysines were identified, among them 1494 lysines were from cell surface annotated proteins. **b)** GOCC analysis of the labeled proteins in HEK293T cells (top) and Hela cells (bottom).

Minor comments:

6. The authors design their probe (OPA-SS-alkyne) to preferentially label cell surface proteins by incorporation of a disulfide linkage that is hypothesized to be reduced under

cytoplasmic conditions. This design should be evaluated further by demonstrating that this disulfide is in fact reduced under conditions that mimic the cytoplasm.

Response: We thank the reviewer's comments. This is the similar question to 4) from reviewer 2. We investigated the concentration of GSH on efficiency of disulfide cleavage in the linker. As shown in Figure R4, the fluorescence intensity decreased with the increasing concentration of GSH, indicating the GSH could partly cleave the disulfide. Therefore, even if the probe enters the cell, the protein in the cell will not be captured during the following purification step.

7. The authors should comment on whether OPA-SS-alkyne reacts with protein N-termini. Is this seen in proteomics experiments? If this is unclear, it can be tested with a peptide with a free terminus without a Lysine sidechain

Response: Thanks for the suggestion. OPA does react with the unmodified N-termini according to the previous report.⁴ However, the N termini of proteins are exposed to a highly diverse set of modifications⁵. For example, about 76% of HeLa proteins are fully acetylated at their N terminus.⁶ In view of the much lower abundance of free N-terminal compared to lysine, the reaction with free N-terminal may show little effect on the proteomics experiments. We also used PEAKS ONLINE to search the raw MS data of 100 uM OPA-S-S-alkyne label proteomics experiments with OPA-S-S-alkyne modification on N-terminal added. Only 42 peptides are considered to be modified by OPA-S-S-alkyne on N-terminal, whereas 3942 peptides are modified by OPA-S-S-alkyne on lysine. As a result, the reaction with free N-terminal has little effect on quantifying the lysine reactivity.

8. On page 7, the paragraph starting "Except common drug target and ligand..." doesn't make sense. The authors should revise this sentence.

Response: We thank the reviewer for pointing out the problem and the related sentence has been corrected in the revised manuscript.

9. In experiments in which HEK293T cells are transfected with ENG and BMP9 the

authors should comment on whether HEK293T cells endogenously express one or both proteins. This factor could influence the findings in this cell line.

Response: We thank the reviewer for this suggestion. According to the previous study (<https://www.thermofisher.cn/cn/zh/antibody/product/CD105-Antibody-clone-3A9-Monoclonal/MA5-17041>) and our data towards HEK293T, HEK293T cells do not endogenously express ENG. So, the expression of ENG in HEK293T cells is induced by our transfection. BMP9 is a secreted protein, also present in serum, with a concentration of approximately 10 ng/ml.⁷ To minimize potential interference with experimental results, we previously subjected cells to serum starvation, then labeled them with OPA-S-S-alkyne and subsequently treated them with an excess of BMP9 (25 ng/ml) with incubation on ice. Therefore, we believe that these factors had little influence on the results.

10. On page 7 the text says “nearly half (108/224) of the highly reactive lysines were annotated to be PTM sites...” How can these lysines be labeled if they are already modified with PTMs? Further explanation is needed

Response: Thanks for the reviewer’s comment. The text that “nearly half (108/224) of the highly reactive lysines were annotated to be PTM sites...” is based on the newly published protein lysine modifications database,⁸ which means that these lysines has the potential to be modified. However, protein translational modification is a dynamic process, depending on the changing conditions.⁹ And modifications serve the important purpose of increasing a protein's functional diversity by altering its basic physical and chemical properties such as its structure, activity, stability, cellular localization, and interaction profile.¹⁰ As a result, targeting these potential PTM sites may help regulate related cellular process.

11. Complementary enzymatic methods to measure protein N-termini exposed on the cell surface should be mentioned and discussed (<https://www.pnas.org/doi/10.1073/pnas.2018809118>)

*Affinity-based methods for labeling specific N-terminal amines

(<https://onlinelibrary.wiley.com/doi/full/10.1002/anie.201403214>) or lysine residues (<https://pubs.acs.org/doi/full/10.1021/acs.bioconjchem.2c00334>) should be discussed for context.

Response: Thanks for the advice. Discussion about these research has been added in the introduction.

12. The word “obvious” is used frequently throughout the text. Many of the assertions did not seem obvious to me.

Response: We thank the reviewer for the advice. Such statement was deleted in the revised manuscript.

1. Zhang, Q. *et al.* OPA-Based Bifunctional Linker for Protein Labeling and Profiling. *Biochemistry* **59**, 175–178 (2020).
2. Chia, C.P. & Luna, E.J. Phagocytosis in *Dictyostelium discoideum* is inhibited by antibodies directed primarily against common carbohydrate epitopes of a major cell-surface plasma membrane glycoprotein. *Experimental cell research* **181**, 11–26 (1989).
3. Ward, C.C., Kleinman, J.I. & Nomura, D.K. NHS-Esters As Versatile Reactivity-Based Probes for Mapping Proteome-Wide Ligandable Hotspots. *ACS chemical biology* **12**, 1478–1483 (2017).
4. Tung, C.L., Wong, C.T., Fung, E.Y. & Li, X. Traceless and Chemoselective Amine Bioconjugation via Phthalimidine Formation in Native Protein Modification. *Organic Letters* **18**, 2600–2603 (2016).
5. Millar, A.H. *et al.* The Scope, Functions, and Dynamics of Posttranslational Protein Modifications. *Annual review of plant biology* **70**, 119–151 (2019).
6. Arnesen, T. *et al.* Proteomics analyses reveal the evolutionary conservation and divergence of N-terminal acetyltransferases from yeast and humans. *Proceedings of the National Academy of Sciences of the United States of America* **106**, 8157–8162 (2009).
7. van Caam, A. *et al.* The high affinity ALK1-ligand BMP9 induces a hypertrophy-like state in chondrocytes that is antagonized by TGF β 1. *Osteoarthritis and cartilage* **23**, 985–995 (2015).
8. Zhang, W. *et al.* CPLM 4.0: an updated database with rich annotations for protein lysine modifications. *Nucleic acids research* **50**, D451–d459 (2022).
9. Seet, B.T., Dikic, I., Zhou, M.M. & Pawson, T. Reading protein modifications with interaction domains. *Nature reviews. Molecular cell biology* **7**, 473–483 (2006).

10. Aye-Han, N.N., Ni, Q. & Zhang, J. Fluorescent biosensors for real-time tracking of post-translational modification dynamics. *Current opinion in chemical biology* **13**, 392-397 (2009).

Reviewers' Comments:

Reviewer #2:

Remarks to the Author:

The authors have adequately addressed this Reviewer's comments.

Reviewer #3:

Remarks to the Author:

The authors have responded to all comments and have provided useful new data. However, there are still a few major issues (listed below) and insufficient responses to specific comments that must be addressed prior to publication.

Reviewer major comments:

*It seems that many of the figures presented in revision responses are not incorporated into the revised manuscript (Figures R1-R6). The purpose of reviewer comments is to improve the manuscript not to correspond with the authors directly. Unless there is a very compelling reason for this omission, this should be changed to incorporate all new data described in revisions into the manuscript/SI (with corresponding new text). This needs to be addressed before the manuscript can be accepted.

The full gel/blot for Figure R4 and Figure S17 should be included in source data.

The authors should ensure that every gel/blot has an uncropped version available.

Follow up to previous comments and responses

Original comment 2. The assay used to measure cell viability (Figure 2d), covers a very short time frame. Although cells are only exposed to labeling agent for a short duration it is possible that they are exerting toxic effects on cells that will not be obvious in cell viability assays after only 10 minutes of measurement. The claim that the labeling technique is non-toxic would be strengthened by alternative assays that measure membrane integrity, cell viability over longer durations, or membrane trafficking assays (transferrin receptor internalization).

Original Response: We are sorry for the inaccurate statement. The processing time of 10 minutes is based on the time for efficient labeling and the label time in the following SILAC-ABPP experiments. We agree that the claim of non-toxic labeling technique is inaccurate. Related statement has been revised to "The cell viability has no significant change after 10 min treatment of the probe".

Reviewer follow up: The authors must show that this treatment does not induce membrane

permeation (which could lead to erroneous assignment of extracellularly accessible proteins) in treated cells under these conditions. This can be achieved using a commercially available dye like Sytox Green (<https://www.thermofisher.com/us/en/home/life-science/cell-analysis/fluorophores/sytox-green-stain.html>) applied after labeling with ABPP treatment.

Original Reviewer Comment 4. On page 7 the text says, "The confocal fluorescence images of K285R mutant also showed diminished colocalization between BMP9 and ENG (Figure S17), which proved that K285 also has an obvious influence on the colocalization of BMP9 and ENG." Fluorescence colocalization is an imprecise and somewhat unreliable method for measuring interactions of cell surface proteins. This is particularly true after fixing cells and only showing one data set with a few cells per field of view. Another confusing aspect of these images is that it appears that BMP9 is mostly cytoplasmic in WT ENG conditions (but not other conditions). The claim that BMP9-ENG interactions are disrupted upon labeling is not convincing and the authors should consider other methods to strengthen these claims. Also, the phrase "proved" should be avoided in scientific writing.

Original Response: We are really grateful to the reviewer for the suggestion. We revised the manuscript accordingly. Regarding the phenomenon of BMP9 is mostly cytoplasmic in WT ENG conditions, we hypothesize that this is due to the interaction between BMP9 and WT ENG, forming a complex that enters the cell, while the interaction between BMP9 and KR ENG is relatively weak. As BMP9 is a secreted protein present in serum, approximately 10 ng/ml,⁷ we did not supplement additional BMP9 or induce serum starvation in our immunofluorescence experiments. This was done to observe the differences in BMP9 and ENG under untreated conditions in both cell types. Therefore, the cytoplasmic BMP9 detected in these experiments may originate from endogenous or serum sources. In the WT condition, it still interacts with ENG, leading to the detection of a protein complex in the cytoplasm. We also quantified the BMP9 without the label of OPA-S-S-alkyne (Figure S16 and R6), and observed that the amount of BMP9 were lower in the K285R mutant compared to the wild type (WT), which is consistent with the immunofluorescence.

Reviewer follow up: Details for the antibody used for BMP9 staining should be listed in the methods section and not only the reporting summary.

The explanation for the ENG-mediated import of BMP9 makes sense but this information needs to be shared in the manuscript text to provide context for interpreting the results presented. It was not clear from experimental descriptions that serum starvation was included to avoid the presence of bovine serum BMP9.

Looking at colocalization coefficients for a single cell is not enough to make a robust conclusion that two proteins are co-localized. As mentioned in the first round of comments, the authors should either present stronger evidence (multiple cells in different fields analyzed by this method or other methods) or remove this claim of colocalization.

Reviewer #2 (Remarks to the Author):

The authors have adequately addressed this Reviewer's comments.

Response: Thank you very much for your time.

Reviewer #3 (Remarks to the Author):

The authors have responded to all comments and have provided useful new data. However, there are still a few major issues (listed below) and insufficient responses to specific comments that must be addressed prior to publication.

Reviewer major comments:

1) It seems that many of the figures presented in revision responses are not incorporated into the revised manuscript (Figures R1-R6). The purpose of reviewer comments is to improve the manuscript not to correspond with the authors directly. Unless there is a very compelling reason for this omission, this should be changed to incorporate all new data described in revisions into the manuscript/SI (with corresponding new text). This needs to be addressed before the manuscript can be accepted.

Response: We thank the reviewer for pointing it out. New data in response has been added in the revised version.

2) The full gel/blot for Figure R4 and Figure S17 should be included in source data.

The authors should ensure that every gel/blot has an uncropped version available.

Response: We apologize for our negligence and related data has been added in the revised version.

Follow up to previous comments and responses

1) Original comment 2. The assay used to measure cell viability (Figure 2d), covers a very short time frame. Although cells are only exposed to labeling agent for a short duration it is possible that they are exerting toxic effects on cells that will not be obvious in cell viability assays after only 10 minutes of measurement. The claim that the labeling technique is non-toxic would be strengthened by alternative assays that measure membrane integrity, cell viability over longer durations, or membrane trafficking assays (transferrin receptor internalization).

Original Response: We are sorry for the inaccurate statement. The processing time of 10 minutes is based on the time for efficient labeling and the label time in the following SILAC-ABPP experiments. We agree that the claim of non-toxic labeling technique is inaccurate. Related statement has been revised to "The cell viability has no significant change after 10 min treatment of the probe".

Reviewer follow up: The authors must show that this treatment does not induce membrane permeation (which could lead to erroneous assignment of extracellularly accessible proteins) in treated cells under these conditions. This can be achieved using a commercially available dye like Sytox Green (<https://www.thermofisher.com/us/en/home/life-science/cell-analysis/fluorophores/sytox-green-stain.html>) applied after labeling with ABPP treatment.

Response follow up: Thank you for the advice. We tested the membrane permeation of cells treated with different concentrations of OPA-S-S-alkyne for 10 min and found that OPA-S-S-alkyne treatment had little effect on the membrane permeability of the cells (Figure R1). Related data is added to the revised manuscript.

Figure R1. Cells were treated with cold methanol or PBS or different concentrations of OPA-S-S-alkyne for 10 min and incubated with STYOX Green for 30 min, followed by confocal fluorescent microscopy analysis.

2) Original Reviewer Comment 4. On page 7 the text says, “The confocal fluorescence images of K285R mutant also showed diminished colocalization between BMP9 and ENG (Figure S17), which proved that K285 also has an obvious influence on the colocalization of BMP9 and ENG.” Fluorescence colocalization is an imprecise and somewhat unreliable method for measuring interactions of cell surface proteins. This is particularly true after fixing cells and only showing one data set with a few cells per field of view. Another confusing aspect of these images is that it appears that BMP9 is mostly cytoplasmic in WT ENG conditions (but not other conditions). The claim that BMP9-ENG interactions are disrupted upon labeling is not convincing and the authors should consider other methods to strengthen these claims. Also, the phrase “proved” should be avoided in scientific writing.

Original Response: We are really grateful to the reviewer for the suggestion. We revised the manuscript accordingly. Regarding the phenomenon of BMP9 is mostly cytoplasmic in WT ENG conditions, we hypothesize that this is due to the interaction between BMP9 and WT ENG, forming a complex that enters the cell, while the interaction between BMP9 and KR ENG is relatively weak. As BMP9 is a secreted protein present in serum, approximately 10 ng/ml,⁷ we did not supplement additional BMP9 or induce serum starvation in our immunofluorescence experiments. This was done to observe the differences in BMP9 and ENG under untreated conditions in both cell types. Therefore, the cytoplasmic BMP9 detected in these experiments may originate from endogenous or serum sources. In the WT condition, it still interacts with ENG, leading to the detection of a protein complex in the cytoplasm. We also quantified the BMP9 without the label of OPA-S-S-alkyne (Figure S16 and R6), and observed that the amount of BMP9 were lower in the K285R mutant compared to the wild type (WT), which is consistent with the immunofluorescence.

Reviewer follow up: Details for the antibody used for BMP9 staining should be listed in the methods section and not only the reporting summary.

The explanation for the ENG-mediated import of BMP9 makes sense but this information needs to be shared in the manuscript text to provide context for interpreting the results presented. It was not clear from experimental descriptions that serum starvation was included to avoid the presence of bovine serum BMP9.

Looking at colocalization coefficients for a single cell is not enough to make a robust conclusion that two proteins are co-localized. As mentioned in the first round of comments, the authors should either present stronger evidence (multiple cells in different fields analyzed by this method or other methods) or remove this claim of colocalization.

Response follow up: We are grateful to the reviewer for the suggestions. The description towards serum starvation has been added to the revised version. In addition, we removed statement about colocalization because no more data were available to support it.